# Co-Evolution of Opioid and Adrenergic Ligands and Receptors: Shared, Complementary Modules Explain Evolution of Functional Interactions and Suggest Novel Engineering Possibilities

**DOI:** 10.3390/life11111217

**Published:** 2021-11-10

**Authors:** Robert Root-Bernstein, Beth Churchill

**Affiliations:** Department of Physiology, Michigan State University, East Lansing, MI 48824, USA; church49@msu.edu

**Keywords:** receptor–ligand co-evolution, opioid, adrenergic, cross-talk, molecular complementarity, modular, homodimers, heterodimers

## Abstract

Cross-talk between opioid and adrenergic receptors is well-characterized and involves second messenger systems, the formation of receptor heterodimers, and the presence of extracellular allosteric binding regions for the complementary ligand; however, the evolutionary origins of these interactions have not been investigated. We propose that opioid and adrenergic ligands and receptors co-evolved from a common set of modular precursors so that they share binding functions. We demonstrate the plausibility of this hypothesis through a review of experimental evidence for molecularly complementary modules and report unexpected homologies between the two receptor types. Briefly, opioids form homodimers also bind adrenergic compounds; opioids bind to conserved extracellular regions of adrenergic receptors while adrenergic compounds bind to conserved extracellular regions of opioid receptors; opioid-like modules appear in both sets of receptors within key ligand-binding regions. Transmembrane regions associated with homodimerization of each class of receptors are also highly conserved across receptor types and implicated in heterodimerization. This conservation of multiple functional modules suggests opioid–adrenergic ligand and receptor co-evolution and provides mechanisms for explaining the evolution of their crosstalk. These modules also suggest the structure of a primordial receptor, providing clues for engineering receptor functions.

## 1. Introduction

Cross-talk between opioid receptors and adrenergic receptors has been well-characterized both in laboratory and clinical studies. Opioid compounds such as morphine and the enkephalins enhance adrenergic receptor activity [1,2,3,4,5,6,7,8,9,10,11,12,13,14,15], while adrenergic compounds such as epinephrine, norepinephrine, and propranolol enhance opioid receptor activity [16,17,18,19,20,21,22,23,24,25,26,27,28,29,30,31]. This enhancement is characterized by several unusual characteristics. One is that, in the presence of each other, opioids and adrenergic drugs produce more than additive effects even though neither ligand can activate the other’s receptor [1,2,3,4,5,6,7,8,9,10,11,12,13,14,15,16,17,18,19,20,21,22,23,24,25,26,27,28,29,30,31]. Enhancement also results in significantly increased duration of receptor activation and prevention of down-regulation of receptor function. In the case of opioids, enhancement by adrenergic drugs results in delaying onset of tolerance [32,33,34,35,36,37], while in the case of adrenergic compounds, enhancement results in the delay of development of the so-called “rebound effect” due to tachyphylaxis and receptor internalization [14,38]. Enhancement can also reverse desensitization of the receptor so that treating tissues that have ceased to respond to opioids with adrenergic compounds can re-institute their sensitivity to opioids [35,36,37], while treating tissues that have ceased to respond to adrenergic compounds with opioid-like compounds can re-institute their adrenergic sensitivity [38]. In all of these cases, the enhancement is mediated by binding to extracellular regions of the receptors [2,39,40,41,42,43,44]. For example, morphine enhances adrenergic activity in isolated heart [45] and atrial preparations [46] in a manner that is not blocked by the opioid antagonist naloxone (which binds to the opioid receptor), demonstrating that the opioid is not working by activation of its own receptor but must alter the activity of the adrenergic receptor. Other research clearly indicates that opioid and adrenergic receptors can form complexes (heterodimers or oligomers) that alter each other’s activity. Mechanisms invoked to explain opioid–adrenergic receptor crosstalk include the formation of receptor heterodimers [47,48,49,50,51,52,53,54,55], interactions between second messenger systems of the receptors [56,57,58], and the presence of extracellular allosteric binding regions for the other class of ligand [2,39,40,41,42,43,44]. Each of these mechanisms poses interesting evolutionary problems that we address in this paper. How, for example, do different types of receptors evolve to co-activate each other’s second messenger systems? How could they evolve to form functional heterodimers? How could two different classes of receptors, one presumably for peptides (opioids) and one for monoamines (adrenergics), co-evolve extracellular allosteric regions capable of enhancing receptor activity upon binding of the other class of ligands? Moreover, if more than one of these mechanisms are at work, what evolutionary selection factors would lead both receptor types to incorporate multiple, interactive mechanisms into their functions in tandem? While such questions may seem very theoretical, their answers may have very practical applications for drug design and for engineering receptors for various uses such as drug screening or sensing.

While it is possible, but extremely unlikely, that adrenergic and opioid receptors had separate origins and then underwent convergent evolution that increasingly integrated their functions, it is difficult to conceive of random evolutionary processes and selection mechanisms that would not only permit the receptors themselves to form functionally integrated heterodimers but also lead to each receptor evolving allosteric binding sites for the other’s ligand. We propose that a simpler solution to these puzzles is that the adrenergic and opioid receptors co-evolved from a common evolutionary origin. Structure and function were integrated from the outset, and receptor specificity evolved as a secondary result. However, rather than proposing a typical evolutionary tree-like process in which the receptors evolved from a common ancestor by means of selection on an accumulation of single nucleotide polymorphisms, we propose that the evolution of the opioid and adrenergic receptors occurred through the swapping and rearranging of a shared set of common modules that were selected for their molecular complementarity. Typical evolution through the accumulation of polymorphisms then modified and honed the resulting structures producing the variety of opioid and adrenergic receptors that now characterize vertebrate organisms but retained their shared modular basis.

In order to make our argument, we have structured our paper in an unusual form that does not follow the typical introduction, methods, results, and discussion format. Instead, we combine four elements in a synthetic manner, introducing review material and methods where appropriate to the development and testing of our hypothesis. The first element is a general theory of receptor–ligand co-evolution based on modular complementarity and its specific implications for understanding adrenergic and opioid ligand and receptor cross-talk. The second is a review of existing evidence concerning modular complementarity as it relates to adrenergic and opioid ligand co-evolution. The third element consists of novel tests of modular complementarity as it applies to co-evolution and crosstalk in adrenergic and opioid receptors. The final element consists of a series of experimentally testable, logical predictions that follow uniquely from the previous three elements involving the artificial evolution and the engineering of novel receptor–ligand functions. 

## 2. General Theory of Receptor–Ligand Evolution

One of the most perplexing problems posed by the evolution of any receptor is how it co-evolved with its ligands. This problem, as noted above, is exacerbated when trying to figure out how two different types of receptors co-evolved to be able to form functional heterodimers and becomes even more perplexing when each receptor responds allosterically to the other’s ligands. Given that a peptide ligand that is just ten amino acids in length could have any of 10^20^ sequences, and a potential receptor protein for that ligand that is 400 amino acids in length could have any of 400^20^ possible sequences, how could a specific ligand end up “finding” its corresponding receptor within “protein space” and then instantiating the pairing as a viable, genetically encoded set? The problem would appear to become exponentially more difficult when trying to account for the co-evolution of two sets of receptors with two sets of ligands, and still more difficult when the two types of ligands bind to both types of receptors and the receptors are able to bind to each other, altering each other’s functions. The number of permutations that would have to be explored seems to be unreasonably large.

Some years ago, Dwyer ([59] and this Special Issue) proposed a clever solution to the basic receptor–ligand evolution problem by suggesting that ligands and receptors evolved from self-complementary peptides, identifiable by their ability to bind to themselves. One of the pairs evolved into the ligand; the other was elaborated into the receptor. Thus, only one short genetic sequence would have been needed to trigger the process, and the receptor could then have evolved by gene duplication followed by ligation to membrane-spanning and second-messenger-activating sequences [60] (Figure 1). In this way, binding specificity was built into the evolutionary basis of the ligand–receptor pair from the beginning by the homo-complementarity of the peptide ligand itself. Dwyer prototyped his theory by demonstrating experimentally the binding of the self-aggregating peptide bungarotoxin, an acetylcholine antagonist, to the acetylcholine receptor and then demonstrating that the bungarotoxin binds to one or more highly homologous sequences in the receptor. These results have been independently confirmed and the participation of the bungarotoxin-like receptor sequences in acetylcholine binding demonstrated [61,62].

Subsequently, Root-Bernstein proposed that Dwyer’s theory could be expanded to include hetero-complementary pairs in which two different complementary peptides were involved or in which only one of the two molecules was a peptide. One peptide component of the pair evolved into the receptor, while the other component (the complementary peptide or non-peptide) remained the ligand. This extension permitted Dwyer’s concept to be expanded to explain the evolution of receptors for hetero-complementary peptide pairs and for non-peptide ligands such as the monoamines [60,63]. The application of the extended theory to hetero-complementary peptide pairs was made to insulin and glucagon (which bind to each other with high affinity but have very different sequences), the receptors of which each contain homologous copies of the other peptide; these homologous sequences are found in ligand-binding regions of the receptors [63,64,65]. The extended theory was also applied to, and experimentally tested, on the evolution of glucose transporters from insulin-like modules based on the fact that the transporter consists of multiple insulin-like domains, and insulin itself has several glucose binding sites [64,66].

Dwyer has subsequently extended his work to identify shared adhesion binding sites among interleukins and gastrointestinal peptides [67], while Sadanandam et al. used a combination of the Dwyer and Root-Bernstein approaches to identify, and then experimentally demonstrate, Plexin B3 and Neuropilin-2 as novel binding partners for Semaphorin 5A [68]. 

This paper extends these previous results to the opioid and adrenergic receptor classes within a broader theoretical framework concerning the evolution of interactomes as the basis for integrated physiological systems. Complementary pairings between molecules would have been selected during the origins of life because their complexes protect the components from degradative processes so that they survive longer than the individual components; the complexes may have novel properties, and binding provides a basis for building functional interactions [69,70,71]. It follows that molecularly complementary compounds (i.e., those that bind to each other) will be very likely to alter each other’s physiological activity, and, conversely, compounds that alter each other’s physiological activity are likely to be molecularly complementary [69,72,73,74]. Additionally, a “molecular paleontology” should leave traces of the origins of these prebiotic selection criteria in modern molecular complexes and their functions [69,70,71]. These general principles can be applied to make specific predictions regarding opioid–adrenergic interactions that are summarized below in Figure 2:(1)Since adrenergic compounds modify the effects of opioid compounds and vice versa (see above), adrenergic compounds should be molecularly complementary to opioid compounds, which is to say that adrenergic compounds should bind specifically to opioids.(2)Opioid peptides that bind adrenergic compounds (hetero-complementarity) should provide the basis for the molecularly complementary modules upon which evolution has built receptors and transporters for adrenergic compounds so that adrenergic receptors should have opioid-like modules within their ligand binding regions.(3)Self-aggregation (homo-complementarity) of opioid peptides should provide the basis for the molecularly complementary modules which evolution has built receptors and transporters for opioid compounds.(4)Some of these opioid-like regions of opioid receptors should bind adrenergic compounds (heterocomplementarity) acting as allosteric modifiers of opioid receptor function.(5)Similarly, some of the opioid-like regions of adrenergic receptors should bind opioids (homocomplementarity again) and act as allosteric modulators of the receptor.

In other words, Predictions 1 through 5 state that the molecular complementarity of the ligands became embedded in the receptors through both homo- and hetero-complementarity such that the receptors retain some affinity for both ligands.

(6)Opioid and adrenergic receptors will therefore be likely to share common, evolutionarily conserved modules.(7)Because opioid and adrenergic receptors can homodimerize as well as heterodimerize with each other, these conserved modules are likely to include some transmembrane regions of the receptors involved in receptor oligomerization.(8)The sum of these shared modules may help to explain how these receptors have integrated functions that result in mutual enhancement of function and how they manage their cross-talk.

Each of these propositions is tested below using combinations of previously published results as well as new data from proteonomic homology methods.

## 3. Literature Review of Results Relevant to Theoretical Predictions

Some of the predictions made above have already been experimentally investigated. We consider these results in terms of the hypothesis that adrenergic and opioid receptors and their ligands co-evolved by means of conserved, molecularly complementary modules.

**Prediction** **1.***Adrenergic compounds should be molecularly complementary to opioid compounds since each molecular species alters the physiological effects of the other*.

As noted in the Introduction, combining adrenergic and opioid compounds alters the effects of each on its respective receptor. As predicted, adrenergic compounds do bind to opioids and vice versa. Nuclear magnetic resonance spectroscopy, ultraviolet spectroscopy, and capillary electrophoresis have been employed to demonstrate that compounds such as epinephrine, norepinephrine, amphetamine, albuterol, propranolol, etc., bind directly to morphine, methionine- and leucine-enkephalin and endomorphin (Table 1) [17,75,76,77]. Thus, the small-molecule complementarity predicted by Dwyer and Root-Bernstein is present in opioid–adrenergic pairs.

**Prediction** **2.***Adrenergic receptors should have opioid-like modules within their ligand binding regions*.

Since adrenergic compounds bind to opioids, primitive versions of opioid peptides such as pro-enkephalin and the endorphins may have provided the basic modules from which adrenergic receptors evolved. Molecular paleontology therefore predicts that remnants of these opioid-like sequences should be conserved in the binding regions of adrenergic receptors. Indeed, an LALIGN comparison of opioid peptides with adrenergic receptors (methods explained more fully in the Figure 3 caption) reveal that several regions of significant homology exist between endorphins or pro-enkephalins and adrenergic receptors (Figure 3). The location of these regions of homology map to transmembrane (TM) sequences TM5 and extracellular (EC) sequences EC3 that form part of the ligand-binding pocket of the receptors according to the UniProtKB website. The locations of these homologies within the overall receptor sequences are plotted below in Prediction 6. These homologies are statistically significant as will be discussed in greater detail in the statistical tests section that follows the final prediction below.

**Prediction** **3.***Self-aggregation (homo-complementarity) of opioid peptides should provide the basis for the evolution of opioid receptors*.

Dwyer’s theory proposes that homo-complementary peptides may evolve into ligand–receptor pairs [59] so that it may be predicted that opioid peptides are homo-complementary. This is the case.

Enkephalins have been found to self-aggregate into homodimers arranged in an anti-parallel beta ribbon conformation [78,79,80,81,82,83]. Notably, the linearly extended conformation required for homodimerization is also the preferred, lowest-energy conformation and the one that best superimposes well on the pharmacophoric groups of morphine [84]. Since endorphins include the enkephalin sequence, if follows that enkephalins can also bind to endorphins, although such aggregation has not apparently been experimentally tested.

Additionally, beta endorphin self-aggregates to form amyloid fibrils [85,86,87,88,89], as does proenkephalin (PENKA) [90], so that both these molecules are homo-complementary and satisfy Dwyer’s criterion for being possible precursors for opioid peptide receptors.

Following Dwyer further, opioid receptors should display evidence of opioid-like sequences associated with their ligand-binding regions. In fact, the beta endorphin and PENKA peptides have significant homologies to one another and to the mu and delta opioid receptors, and these homologies are conserved from fish through human beings (Figure 4), a fact that is explored in additional vertebrate species below (zebrafish are used as an example here because they represent the most evolutionarily divergent species from *Homo sapiens* for which the ranges of opioid and adrenergic receptor and ligand sequences were available). These homologies occur within the opioid ligand binding site of the opioid receptors, as will also be demonstrated in Prediction 6 below, and additional homologies are found between opioid receptors and pro-opiomelanocortin and beta endorphin in the extracellular loops. These extracellular loop homologies are significant both in terms of the fact that opioids often interact transiently with these extracellular loops prior to being drawn into their high-affinity binding site that exists in a pocket formed by the transmembrane sequences of the receptor [2,39,40,41,42,43,44] (see Figure 5). These extracellular homologies are also important with regard to adrenergic binding, as is discussed in the next prediction. Additionally, these homologies are statistically significant, as will be discussed in greater detail in the statistical tests section the follows the final prediction below.

**Prediction** **4.***Some opioid-like regions of opioid receptors should bind adrenergic compounds acting as allosteric modifiers of opioid receptor function*.

Since adrenergic agonists and antagonists bind directly to opioid peptides, and opioid-peptide-like regions are found within the opioid binding site of the opioid receptor as well as in its extracellular regions, it follows adrenergic compounds should bind to opioid receptors within one or more regions that are homologous to opioid peptides. This appears to be the case.

To begin with, adrenergic agonists and antagonists do bind directly to opioid receptors blocking access to the high-affinity opioid binding site. Among adrenergic compounds found to be effective competitors for naloxone binding to opioid receptors are: clonidine and L-phenylephrine, alpha adrenergic agonists; phentolamine, an alpha 1- and alpha-2 adrenergic antagonist; prazosin, an alpha 1-adrenergic antagonist; yohimbine, an alpha 2-adrenergic antagonist [91]. Additionally, xylazine, an analogue of clonidine and an agonist at the α2 class of adrenergic receptor, as well as the adrenergic neurotoxin DSP4, also bind directly to opioid receptors with an IC50 competing with naloxone of about 1 µM [92,93]. Beta-adrenergic antagonists competitively bind to opioid receptors as well, displacing the enkephalin agonist [3H]D-Ala2-Met5-enkephalinamide and the morphine antagonist [3H]naloxone [94]. Correspondingly, the alpha-adrenergic antagonists phentolamine and phenoxybenzamine at about 1 micromolar blocked 50% of 5 nanomolar naloxone binding to opioid receptors in brain tissue. This was ten times more potent than the IC50 of codeine and about ten times less potent than the IC50 of methadone [95,96,97].

Epinephrine binds directly to purified mu opioid receptor (MOR) with about the same affinity as methadone [16,41,77] and enhances binding of morphine to the receptor by a factor of about ten [41]. Binding studies to MOR peptide fragments suggest that the epinephrine binding site on MOR is most likely located in a pocket defined by the juxtaposition of the second and third extracellular loops of the receptor (Table 2) though, notably, the first extracellular region also binds epinephrine [16,41,77]. Each of these epinephrine-binding regions corresponds to a sequence with high homology to either pro-opiomelanocortin (and, more specifically, beta endorphin) or pro-enkephalin (Section 3, above). Thus opioid-peptide-like regions of opioid receptors appear to provide binding sites for adrenergic compounds on opioid receptors.

**Prediction** **5.***Some of the opioid-like regions of adrenergic receptors should bind opioids and act as allosteric modulators of the receptor*.

Just as adrenergic compounds modify opioid receptor function, opioids and opioid antagonists are known to modify adrenergic receptor function (Introduction). Following the principles set out in the Introduction, it can be predicted that adrenergic receptors have allosteric opioid binding sites and that they should be composed of opioid-like regions adapted to specifically bind adrenergic compounds. This also appears to be the case. Studies have demonstrated that naloxone and morphine can enhance adrenergic receptor activity in thoracic aorta, which is a tissue notable for lacking opioid receptor, therefore suggesting direct action of opioids on adrenergic receptors. Indeed, two κ opioid antagonists are known to bind directly to adrenergic receptors: nor-binaltorphimine (nor-BNI), a kappa-selective opioid antagonist bound to the α(2C)-adrenoceptor (K(i) = 630 nM), and 6′-guanidinonaltrindole (6′-GNTI) bound to the α(1A)-adrenoceptor (EC_50_ = 41 nM), enhancing calcium mobilization by noradrenaline. Neither compound directly activated the receptors [2].

Additionally, as summarized above in Table 2, morphine and naloxone bind directly to extracellular loops of the beta 2 adrenergic receptor (regions highly conserved in other adrenergic receptors as well) [44,77]. As with the opioid receptor binding site for adrenergic compounds, the adrenergic receptor binding site for opioids appears to be located at the junction of the second and third extracellular loops (Table 2) [44,77]. The presence of a cysteine residue appears to be of particular importance for the formation of a disulfide bond that keeps the receptor in its highest activity conformation [44,98].

Finally, tramadol and tapentadol, which are generally characterized as being combined mu opioid receptor agonists/norepinephrine reuptake inhibitors, also elicit direct effects on adrenergic receptor function via binding to beta 2 and alpha 2 adrenergic receptors [99,100]. Significant functional effects occur at micromolar concentrations [99,100] and can be blocked by various alpha 2 adrenergic receptor antagonists such as yohimbine and idazoxan [100,101].

Together, Points 4 and 5 demonstrate that many adrenergic receptor ligands bind to and directly affect opioid receptor function while many opioid receptor ligands bind to and directly affect adrenergic receptor function. These affects appear to be mediated in both cases by binding to an allosteric pocket formed at the intersection of the second and third extracellular loops and possibly involving part of transmembrane region 2, all of which are homologous to opioid-peptide-like sequences known to bind either opioids or adrenergic compounds (Figure 5).

**Prediction** **6.***Opioid and adrenergic receptors will therefore share common, evolutionarily conserved modules*.

The previous five predictions lead inexorably to the conclusion that opioid and adrenergic receptors share common modules, many of these associated with peptide opioid sequences. In order to explore this possibility more thoroughly, we compare the sequences of a series of opioid and adrenergic receptors from diverse species utilizing LALIGN (an alignment program for exploring similarities between pairs of proteins). The details of the methods are provided in the Figure 6 caption.

Figure 6 illustrates the high degree of homology between the zebrafish kappa opioid receptor and the zebrafish beta 2 adrenergic receptor. Figure 7 illustrates the high degree of homology between the African clawed toad mu opioid receptor and its alpha 1A adrenergic receptor. Figure 8 illustrates the high degree of homology between the mallard mu opioid receptor and its alpha 1A adrenergic receptor. Figure 9 illustrates the high degree of homology between the mouse kappa opioid receptor and its beta 2 adrenergic receptor. Similar homology studies for the stickleback mu opioid receptor/alpha A1 adrenergic receptor and for the human kappa opioid receptor/beta 2 adrenergic receptor pairs will be discussed separately in Prediction 7 below. Additionally, we have previously published a similar study of the human mu opioid receptor and alpha 1A adrenergic receptor [41].

A number of features are shared by all seven pairs of the homology pairs of the opioid and adrenergic receptors that we studied (zebrafish, clawed toad, and mallard, mouse (Figure 4, Figure 5, Figure 6 and Figure 7) and stickleback, human MOR/alpha ADR, and human KOR/beta ADR (which will be discussed in detail below) and these are summarized in Table 3. Most regions of the opioid and adrenergic receptors are well-conserved across the two classes of receptors with the clear exceptions of the first, third, and fourth extracellular sequences (EC1, EC3, EC4) and the third and fourth intracellular sequences (IC3 and IC4). The rest of the receptors average 60% similarity; four regions—the second extracellular sequence (EC2, the first extracellular loop), transmembrane sequences 2 and 6 (TM2 and TM6), and the first intracellular sequence (IC1, the first intracellular loop)—average 70% similarity or greater across the entire range of species and within both adrenergic and opioid receptors.

The overall impression given by Table 3 that the EC2, TM2, TM6, and IC1 are highly conserved is confirmed by Logoplots of these regions, generated from all fourteen receptor sequences (seven adrenergic and seven opioid receptors of the six species and seven comparisons used to generate Table 3. The method is provided in Figure 10 and applies to Figure 11, Figure 12, Figure 13, Figure 14 and Figure 15 as well. The extracellular domain 2 (EC2), transmembrane region 2, or second helix, (TM2), the transmembrane region (or helix) 6 (TM6), and the first intracellular domain (IC1) Logoplots all reveal the presence of amino acids that are very highly, and in some cases completely, conserved in their positions across all seven comparisons and both types of opioid and adrenergic receptors. While the fourth transmembrane region (TM4) is also well conserved (60%: Table 2), it displays only one completely conserved amino acid (a tryptophan) and a handful of other well-conserved residues, thus demonstrating significantly more divergence over evolutionary time than has occurred in EC2, TM2, TM6, and IC1. It is logical to conclude that shared functions, such as shared affinity for both opioids and adrenergic compounds, is preserved in these conserved regions, and that receptor dimerization is also made possible through conserved sequences in these regions—a point that will be taken up at greater length in the next part.

The second extracellular loop (EC3) provides an additional contrast: it is not well-conserved except at the crucial cysteine position that is essential for maintaining the ligand binding site integrity [103] (Figure 6, Figure 7, Figure 8 and Figure 9). Since EC2 and EC3 make up the main part of the allosteric enhancer binding site (Predictions 4 and 5 above), it follows that the specificity of the enhancer site has evolved in these two regions.

In order to validate the utility of the Logoplot results, the highest-scoring consensus sequences were used to perform a BLAST search against the human proteome and the vertebrate proteome. Since the results of the human and vertebrate searches were virtually identical, only the results of the human proteome search are presented below in Table 4, which demonstrates that the three consensus sequences tested all elicited matches to both OPR and ADR, confirming their shared nature. Notably, other receptor classes were also reliably identified as sharing these consensus sequences as well: among the amines, the dopamine, histamine, and serotonin receptors; and, among the peptide receptors, somatostatin, chemokine, angiotensin II, and neuropeptide B/W and Y. Thus, these consensus sequences are more broadly shared than among just the ADR and OPR. On the other hand, the absence of other classes of GPCR from the homology list in Table 4 also suggests that these ADR-OPR consensus sequences are far from being universally conserved in GPCR generally. The special relationships of the receptors found by this BLAST search of the most highly conserved regions of the ADR and OPR to other GPCR will be investigated further in the Discussion section below.

An equally important observation is that several of the sequences of the highly conserved regions are homologous to the sequences of the opioid precursor proteins pro-enkephalin (PENKA) or pro-opiomelanocortin (PROOPI) (which contains the beta endorphin sequence (BENDO)), as demonstrated in Figure 16, Figure 17 and Figure 18 for stickleback mu opioid receptor versus alpha A1 adrenergic receptor, human mu opioid receptor versus alpha A1 adrenergic receptor, and human kappa opioid receptor versus beta 2 adrenergic receptor. The conservation of these opioid-peptide-like regions across both opioid and adrenergic receptors and across species that span a large portion of the evolutionary tree suggest that these regions represent important functional modules that are resistant to mutation.

Comparing Figure 16, Figure 17 and Figure 18 with each other and with Figure 6, Figure 7, Figure 8 and Figure 9 reveals several important characteristics of the co-evolution of opioid and adrenergic receptors. One is that highly conserved regions within the opioid-adrenergic pairs of one species are highly conserved in all of the other species, a point that is emphasized by the results of the Logoplots for EC2, EC3, TM2, TM4, TM6, and IC1 (Figure 10, Figure 11, Figure 12, Figure 13, Figure 14 and Figure 15). Thus, rather than diverging from one another over evolutionary time, opioid–adrenergic pairs have maintained a high degree of similarity. This conservation argues for the critical importance of these sequences to both OPR and ADR function. Equally important from the perspective of modular complementarity, the similarities between opioid peptides and specific regions of both opioid and adrenergic receptors are also conserved across evolutionarily divergent species (Figure 16, Figure 17 and Figure 18). Regions of the stickleback receptors that mimic opioid peptides are regions in the human opioid and adrenergic receptors that also mimic opioid peptides. Thus, these highly conserved functional regions also tend to be related to the opioid–peptide modules identified above. Additionally, Figure 16, Figure 17 and Figure 18 clearly emphasize the point made by Figure 3, Figure 4 and Figure 5 that these opioid peptide–receptor similarities occur at multiple sites in the receptors and that these similar regions occur both in the extracellular (including the first extracellular sequence, EC1, as well as EC3) and the fourth intracellular region (IC4) of the opioid receptors but only in one extracellular loop (EC3) and the third intracellular region (IC3) of the adrenergic receptors. Thus, these modules appear to have been rearranged in adrenergic receptors as compared with opioid receptors, providing a mechanism for explaining their divergent ligand specificities. This rearrangement is easily observed in the abstracted diagram below (Figure 19) where it is clear that the opioid receptors are characterized by copies of opioid peptides in both their first extracellular and fourth intracellular regions that are not present in the adrenergic receptors, which display opioid receptors in regions that share little sequence similarity to the opioid receptors. The presence of opioid peptide-like regions on the extracellular and intracellular regions of the opioid receptor has possible importance in understanding the evolution of receptor signaling that will be taken up further below.

**Prediction** **7.***Conserved modules include the transmembrane regions of the receptors*.

Our seventh prediction is that homo- and hetero-dimerization domains should be highly conserved between the adrenergic and opioid receptors, thus representing yet another class of evolutionarily stable, homo-complementary modules. Adrenergic receptors can homodimerize and heterodimerize with each other (reviewed briefly below); opioid receptors can homo- and heterodimerize with each other (reviewed briefly below); and adrenergic receptors can heterodimerize with opioid receptors [47,48,49,50,51,52,53,54,55]. Logically, this conservation of dimerization potential requires that the dimerization determinants in adrenergic and opioid receptors be highly conserved as well. Since dimerization regions generally involve transmembrane sequences (discussed in detail below), transmembrane regions involved in dimerization might therefore be expected to be highly conserved, as is summarized in Table 3 and Figure 19. As with the previous predictions, this appears to be the case. As noted in the previous section, all of the transmembrane (TM) regions are conserved to at least 60% across all fourteen of the receptor sequences investigated with the exception of TM5 (57% homology); TM2 and TM6 are particularly well-conserved across all of the opioid and adrenergic receptors (72% and 76%, respectively) (Table 3; Figure 11, Figure 12 and Figure 14). This conservation is also apparent in Figure 6, Figure 7, Figure 8, Figure 9, Figure 16, Figure 17 and Figure 18 and is independent of the opioid–peptide modules.

The question now becomes whether these highly conserved TM regions are particularly important determinants of receptor dimerization. This appears to be the case, though much uncertainty still attains to the specific interactions governing dimerization and even which transmembrane regions of the receptors are involved in both ADR and OPR.

Three very different models have been proposed for homodimerization in beta-2 adrenergic receptors (B2AR), one involving helices V and VI (TM5 and TM6) [104], another involving helix VI (TM6) binding to a complementary helix [105], and the third involving binding between either helices 1 and VII (TM1 and TM7) [106] or between helices I/II with III/IV (TM1/TM2 with TM3/TM4) [107]. Further studies determined that helix I (TM1) was unlikely to be involved in homodimerization since mutations of helix 1 had little effect on dimerization [108]. However, studies of the crystal structure of B1AR also suggested that oligomerization may involve TM1/TM2 interactions as well as TM4/TM5 interactions [109], though the extent to which crystal interactions predict membrane interactions is open to question since the receptors in a crystal are packed in ways that are not evident when they are embedded in a cell membrane. Thus, evidence from studies of alpha-adrenergic receptor homodimerization have demonstrated that neither helix 1 (TM1) nor helix 7 (TM7) are involved, but isolated helices III, IV, V, and VI could each, independently, bind to full-length receptor [110]. Again, the relevance of these studies to membrane-embedded receptor is questionable since these studies were done in solution. Additionally, a peptide derived from helix VI (TM6) was able to inhibit B2AR homodimerization, indicating that TM6 plays an essential role [105]. Finally, TM4 is able to self-associate but not at a sufficiently high affinity to account for observed dimerization interactions [110,111]. In sum, for adrenergic receptors, helix II (TM2) may interact with helices III or IV (TM3 or TM4), while helices V or VI (TM5 or TM6) may also participate in adrenergic receptor dimerization. There is no evidence that helix VII (TM7) is involved in dimerization, and evidence for the participation of TM1 is contradictory.

The situation regarding opioid receptors is equally complex. Crystal structures of the mu opioid receptor (MOR) and kappa opioid receptor (KOR) show closely packed interfaces between TM5 and TM6 as well as a less compact interface involving TM1, TM2, and a helix found in the intracellular domain 4 (IC4) [112]. Again, it is questionable whether crystal interactions are a good indicator of membrane interactions. Notably, MOR has a functional truncated splice version that lacks the first helix (TM1) but which still forms homodimers [113] as well as heterodimers with B2AR [114]. Thus, while helix 1 (TM1) may normally be involved in dimerization, it cannot be necessary for homo- or heterodimerization of MOR. As in adrenergic receptors, TM4 self-aggregates and may also bind to TM5 in DOR homodimerization [115]. Crystal studies demonstrate that MOR oligomerization is mediated by TM5/TM6 binding [116]. Modeling studies of opioid receptor homo- and heterodimerization tend to confirm that TM1/TM2 and TM5/TM6 participate in dimerization. One study revealed high probabilities of interactions between TM1,2 and TM5,6, but also TM4 and TM5 (as well as TM4,5/TM1,2 for heterodimers); TM3 and TM7 were never involved [117]. In opioid receptor heterodimers, additional computer simulations identified TM1/TM2/IC4 complexes and TM3/TM6 complexes as the most likely means of dimerization [118]. So, the consensus of the existing data is that TM1/TM2 may interact with TM5/TM6, while TM 3 and TM7 are never implicated in dimerization, and TM1 and TM4 seem to be dispensable.

In sum, the question of how homo- and heterodimerization are mediated is still open for both adrenergic and opioid receptors. The only firm conclusions that can be reached are that TM1 may participate, but is not necessary, and there is little or no evidence at present for the participation of TM3 or TM7. Notably, TM2, TM4, and TM6 are the best-conserved helical (or transmembrane) regions of the opioid and adrenergic receptors across species (Table 3), suggesting that dimerization may be mediated mostly through interactions involving these helices. Treating each of these sequences as a conserved module may permit novel experiments to be performed to determine which modules are complementary to the others (or to themselves). Additionally, since receptors are often localized in lipid rafts with specific lipid and protein compositions, the possibility that conserved TM regions may have evolved in part due to raft composition should also be considered.

### Statistical Significance of the Results Reported Above

The statistical significance of the homology results reported in the previous four sections can be evaluated in three ways. One is to compare the present results with a previous study of the probability of any given peptide ligand mimicking its own, or any other, receptor [63]. In the study just cited, 13 peptide ligands were tested against 25 peptide receptors (for a total of 325 permutations) using LALIGN to determine how often sequence matches occurred at any given statistical value. Two criteria were employed for significance: one was whether the match included a sequence of 10 amino acids in which at least 5 were identical, and the second was a Waterman–Eggert score of 35 or greater. Only 3.6% of the matches satisfied those simultaneous criteria (i.e., *p* = 0.036), indicating that the probability of any given peptide ligand having a homologous region in a random peptide receptor was extremely small. Of the 17 homologies illustrated in Figure 3 and Figure 4, comparisons of opioid peptide sequences with adrenergic and opioid receptor sequences using LALIGN yielded Waterman–Eggert (WE) scores above 35 (and as high as 57) with corresponding E values (which correspond to internal probability values) of less than one (and as small as 0.001). The higher the Waterman–Eggert score, the less probable the match is to have occurred by chance; conversely, the smaller the E value, the less the probability [119]. The probability that all seventeen homologies illustrated in Figure 3 and Figure 4 would satisfy both WE and E value criteria simultaneously is 0.036/17 or 0.0021.

However, Waterman and Eggert caution that their probability scores should always be complemented with a statistical approach that specifically tests the hypothesis generating the homology search [119], so a second statistical approach involves interpreting the results in terms of a random search of endorphins and proenkephalin sequences with each other and with a range of opioid and adrenergic receptors from a variety of species (Appendix A). These results confirm that the types of high Waterman–Eggert score/low E-values/high identity sequences found in Figure 3 and Figure 4 occur almost solely between evolutionarily related molecules such as endorphins from different species or endorphins with pro-enkephalins. The fact that endorphin- and proenkephalin-like sequences occur within both opioid receptors and adrenergic receptors at equally high Waterman–Eggert scores and E values is therefore indicative of an evolutionary relationship.

Finally, the Waterman–Eggert scores and E values provided for receptor homologies (Figure 6, Figure 7, Figure 8, Figure 9, Figure 16, Figure 17 and Figure 18) as a function of the LALIGN search algorithm also provide internal tests of probability. Comparisons of opioid with adrenergic receptors universally displayed Waterman–Eggert scores or their equivalent Z scores above 200, while the E values range from 10^−24^ to 10^−110^, which are extraordinarily unlikely to occur by chance and are generally found only for evolutionarily related proteins, as is evident from Appendix B.

In short, whatever statistical measure is used, the results reported here are extremely unlikely to have occurred by chance and strongly suggest the non-random conservation of key structural and functional modules across all of the adrenergic and opioid receptors tested.

## 4. Discussion

To summarize, we have demonstrated here that adrenergic and opioid compounds are, as their complementary physiological activities predict, molecularly complementary. Adrenergic compounds bind to extracellular loop peptides derived from the mu opioid receptor while opioid compounds bind to extracellular loop peptides derived from adrenergic receptors. Thus, the complementarity between opioids and adrenergic compounds for each other extends to binding to each other’s receptors. In addition, endorphins and pro-enkephalins are known to self-aggregate (i.e., are self-complementary). Following Dwyer’s proposal that evolution uses complementary modules as the basis for bootstrapping receptors, beta endorphin-, pro-enkephalin-, and/or pro-opiomelanocortin-like sequences are present in the binding sites of opioid receptors. Similarly, following Root-Bernstein’s extension of Dwyer’s theory to hetero-complementary compounds, adrenergic compounds bind to opioids such as morphine and enkephalins and to beta-endorphin- or proenkephalin-like sequences present in adrenergic receptors. Thus, opioid and adrenergic receptors not only share evidence of binding each other’s ligands, but they also share key modules that have the appropriate homo- and hetero-complementarity to make such shared binding possible. These modular regions are mainly located in the first two extracellular loops of the respective receptors and are well-conserved both across evolutionary time and between adrenergic and opioid receptors. Additionally, adrenergic and opioid receptors share some highly conserved transmembrane (helical) regions that appear to play key roles in both homo- and heterodimer (or oligomer) formation. Finally, these shared extracellular and transmembrane modules are also found in a number of other receptor types, including serotonin, melatonin, dopamine, and histamine receptors and neuropeptides B/W, somatostatin, cholecystokinin, and chemokine C-X-C receptors. Thus, a case can be made that molecular complementarity involving conserved modules has played a crucial role in the co-evolution of adrenergic and opioid receptors, as well as other aminergic and peptide receptors, and that these conserved modules supply the molecular mechanisms underlying their heterodimerization, cross-talk, and mutual enhancement.

### 4.1. Evolutionary Models of Results

These results can be considered in light of two very different evolutionary models. One is the well-known model of evolutionary diversification from a common ancestor. It is possible that the adrenergic and opioid receptors did evolve from a common ancestor. Wolf and Grünewald [120] have reported that although one would expect opiate receptors to have evolved from the class of GPCR peptide receptors because they bind enkephalins, various binding site properties of OPR are more similar to aminergic receptors such as the ADR than to peptide receptors. On the other hand, Larhammer et al. [121,122] have found that OPR resemble the neuropeptides B/W receptor more closely than any other receptor. Indeed, in Table 4, we reported that the neuropeptide B/W receptor does share some significant homologies with the Logoplot consensus sequences for TM2 and EC2. Furthermore, an LALIGN global homology search such as those performed here for Figure 4, Figure 5, Figure 6 and Figure 7 yields a Waterman–Eggert score or 778 with E(1) < 1.1 × 10^−58^ for P48146|NPBW2_HUMAN versus P35372|OPRM_HUMAN, which is, indeed, a significantly better set of statistical matches than the ADR-OPR homology results reported above. Moreover, the vast majority of studies of GPCR evolution place adrenergic and opioid receptors onto separate branches of the GPCR evolutionary tree (e.g., [123,124,125,126]): ADR are placed with other amines such as dopamine, histamine, melatonin and serotonin, while OPR are most closely related to somatostatin (the two often classified as being in a single grouping) and, more distantly, with neuropeptide B/W, and other peptide receptors such as the chemokines. One novel aspect of our results is to demonstrate that these segregated sets of receptors actually share highly conserved functional modules.

The differences in results obtained by us and Wolf and Grünewald [120] compared with the other groups are clearly due to analysis of different micro-scale properties by the former and macro-scale or global properties by most other groups. Accepting both results as valid within the parameters of their particular search algorithms, one is faced with various possible interpretations. For example, perhaps OPR and ADR evolved from a common ancestral, aminergic receptor with the somatostating and neuropeptides B/W receptor evolving from the OPR. Or perhaps the OPR evolved from the somatostatin or neuropeptide B/W receptors through a common peptide receptor ancestor. The difficulty with the latter case is that if ADR and OPR evolved from separate branches of the GPCR evolutionary tree (one, an aminergic GPCR, the other, a peptide GPCR), then one is left with the difficulty of explaining how the common elements of ADR and OPR demonstrated here (and by Wolf and Grünewald [120]) emerged and were selected for not only in terms of protein sequence homologies but, more importantly, also for the ability to heterodimerize and for functional crosstalk.

An alternative evolutionary model employs the well-established concepts of gene duplication and modular accretion of proteins. Dwyer [59] and we [60,63,64,65] have suggested that evolutionary processes are highly conservative, re-using adaptive, short homo- and heterocomplementary protein modules by duplicating and swapping them. This modular swapping model does not rely on all receptors having a common ancestor, but rather posits the possibility of building receptors with different specificities out of common protein modules that are mixed and matched in novel permutations through short gene segment duplications and ligations.

A complementary, modular mix-and-match model helps to explain how ligands and receptors co-evolve from short, shared modules rather than accidentally evolving toward increasing specificity from independent origins [59,60,64,65]. The result is a network of commonalities based on short regions of homology rather than a tree of diverging commonalities based on global homologies [63,69]. The data that we have provided here seem to fit such a modular network model. This model would explain how OPR could look simultaneously like other peptide receptors such as the somatostatin, neuropeptides B/W and chemokine receptors (peptide homocomplementarity may be at the root of the specific binding regions of each peptide receptor) [120,121,122,123,124,125,126] and yet share significant homologies with aminergic receptors as well; aminergic receptors seem to have used heterocomplementarity of their ligands for peptides as the basis of their specific binding regions. Such a model also helps to explain how the adrenergic and opioid receptors co-evolved to integrate their mutual functions through both enhanced activity and heterodimerization. Finally, the model appears to be extendable to understanding functional integration through dimerization and enhanced activity to the modularly related receptors listed in Table 4, such as the SST(2A) somatostatin receptor with the mu-opioid receptor [127], the delta-opioid receptor and somatostatin receptor-4 [128], the mu-opioid and serotonin receptors [129,130], dopamine with somatostatin receptors [131], β1- and β2 adrenergic receptors with the somatostatin receptor 5 [132,133], and β1- and β2-adrenergic receptors with adenosine A1 receptors [134]. (Notably, there appear to be no studies of heterodimerization involving neuropeptides B/W receptor, which might provide a robust test of the concepts proposed here, as it can be predicted to dimerize with both aminergic and opioid receptors.)

In addition to the sets of interactions just summarized and documented in this paper, another set is also relevant to understanding adrenergic and opioid receptor co-evolution as well as shared modularity with receptors such as histamine [135]. These interactions involve glutathione (GSH) with amines such as epinephrine and norepinephrine and with opioids such as morphine and enkephalins. Briefly, the primary function of GSH is to recycle oxidized ascorbic acid (dehydroascorbic acid or DHA) back into reduced ascorbic acid (ASC), thereby mediating cellular oxidative stress [136]. GSH also recycles epinephrine, norepinephrine, dopamine, and related adrenergic amines, which oxidize to form toxic adrenochrome-like compounds or forms adducts with them that are then excreted [137,138,139,140,141,142,143,144,145]. Similarly, opioids such as morphine and the enkephalins bind to GSH antagonizing its antioxidant function and forming inactive adducts [146,147,148,149,150,151]. Notably, GSH-like sequences are located in the second or third extracellular loops at the sites of the conserved cysteines in both adrenergic and opioid receptors (Figure 19) where they function like GSH (and with very similar kinetics) to recycle DHA into ASC and to protect adrenergic compounds from oxidation [44,98]. Thus, GSH-like sequences represent yet another molecularly complementary module that is conserved across these receptor classes and across species, thereby functionally linking adrenergic and opioid compounds via a further shared binding motif [152,153]. These GSH-like sequences are also shared by other aminergic receptors including histamine, dopamine, and serotonin [135,152,153,154].

### 4.2. Model of ADR-OPR Co-Evolution Based on Molecularly Complementary Modules

These multiple sets of molecularly complementary interactions between highly conserved modules permit the following schematic model of adrenergic–opioid receptor co-evolution to be proposed based on the previous work of Dwyer [59] and Root-Bernstein [60,63,64,65] (Figure 20). The model assumes eight basic sets of compounds: ascorbic acid (ASC); dehydroascorbic acid (DHA); epinephrine and other adrenergic compounds such as norepinephrine, amphetamine, dopamine and their metabolites (EPI); peptide and non-peptide opioids; glutathione (GSH); glutathione-like polypeptides (GSH-like); and peptides selected for their ability to insert stably into or across cell membranes (transmembrane sequences or TM modules). Early proto-receptors could have evolved from the ligation of gene sequences encoding TM modules with GSH-like modules and/or peptide opioid modules. TM modules were probably selected for their ability to homo-dimerize or homo-oligomerize due to the added stability that such complexes would have conferred on their constituents. Functionality would have been conferred by a series of selection pressures. One selection factor would have been further stabilizing TM complexes by modifying these extended TM modules to have a preferred membrane orientation (which might have been achieved by linking TM modules by means of one or more extra- or intracellular loops, once again carried out by ligation of short, modular genes). Functionalization of such dimers or oligomers might have resulted by selecting for sequences that changed conformation upon binding a ligand at their extracellular face and adding an intracellular sequence that altered conformation or that was released from the cell membrane when extracellular binding occurred. These intracellular sequences would have undergone a second round of selection involving their ability to interact with what we now call second-messenger systems. Some intracellular sequences may have been similar to the extracellular binding sequence so that the “message” conveyed by binding to the extracellular module was also replicated internally. Presumably, the early proto-receptors contained an extracellular binding sequence with a general affinity for signaling molecules that undergo duplication or rearrangement and are then put inside the cell. This “internal image” would explain the presence of intracellular opioid peptide-like sequences corresponding to the extracellular opioid peptide-like sequences in both the adrenergic and opioid receptor classes. Modular combinations that inserted into the membrane in the wrong orientation, that were composed of the wrong number or sequence of modules, or which paired non-functional or misleading “messages”, would have been selected out. The surviving proto-receptors may have evolved better control and specificity by adding additional TM modules attached by additional, more-or-less shared, extracellular or intracellular modules. In the case of the adrenergic and opioid receptors, the extracellular modules share a high degree of homology with GSH-like peptides and opioid peptides such as beta endorphin and pro-enkephalins. While these extracellular modules are similar across both classes of receptors and across evolutionary development, the specificity of the modules for either opioids or adrenergic compounds seems to have diverged either by modular swapping and/or a slow accumulation of mutational events. Both sets of receptors have, however, retained their ability to bind both classes of compounds and to respond allosterically to their combined binding. Moreover, the conservation of TM modules has ensured that the self-aggregation selected for at the outset of proto-receptor evolution has been conserved so that homo-dimerization within each class of receptors is retained as well as the ability to form heterodimers across receptor classes. In short, selection for molecularly complementary complexes, starting with small molecule interactions conserved during mixing-and-matching of shared modules through ligation to form larger peptides and proteins, eventually yielded functionally different functions retaining a high degree of interactivity [60,63].

We reiterate that one of the key (and only partially tested) assumptions underlying the analysis performed here has been that the evolutionary relationship between the opioid and adrenergic classes of receptors is not necessarily through a standard evolutionary tree but rather as the result of rearrangements and swapping of shared modular units. The assumption that receptors evolved by means of modular duplication and swapping was built into Dwyer’s initial conception of peptide receptor evolution. Based on the discovery that many proteins are encoded in genetically dispersed pieces that are assembled during mRNA production (“genes in pieces”) [154], Dwyer proposed that peptide receptors (and protein–protein interactions more generally) evolved by means of duplicating small gene sequences encoding homo-complementary peptides, which were then propagated by transposable exon elements (trexons) to provide ready-made protein-binding modules [67,154]. Such small, mobile exon elements, also called “miniature inverted-repeat transposable elements” or MITES, are ubiquitous and common features of bacterial chromosomes [155,156]. Our results extend Dwyer’s original conception to swapping of complementary modules between classes of receptors and illustrates its application to understanding the multiple ways (mutual ligand binding, allosteric enhancement, and conserved helix interactions) in which opioid and adrenergic receptors are able to interact.

#### 4.2.1. Receptor Evolution as a Case Study in Interactome Emergence

Returning to the broader question of receptor evolution, it is important to stress that these results provide evidence that there are two modes in which novelty has been introduced into proteins, one being the standard model of an accumulation of single nucleotide polymorphisms and a second, more novel one being selection for complementary modules that encode binding motifs. One of the greatest mysteries that the mutation accumulation model leaves unresolved is why the random exploration of permutations does not lead to increasing loss of function over time and ever less integration within living systems when, instead, what is clearly observed across evolutionary time is an ever-increasing tendency toward systems integration exemplified in very stable interactomes. Selection for complementary modules can answer this conundrum. Complementary modules, whether homo- or hetero-complementary, would have been selected because of the increased stability of their aggregates against proteolysis and other causes of degradation or modification [69,70,71]. Their adaptation for uses such as transporter (e.g., [66]) or receptor functions would then have provided another level of favorable selection. Reuse of previously selected complementary modules would then have helped to ensure systems integration as evidence in interactomes. It follows that evolutionary systems evolve by means of a balance between divergent and convergent mechanisms of variation and that evolutionary trees only capture the first type of evolutionary processes. Interactome formation by means of swapping and selection for complementary modules produces, instead, a branched network of connections among evolutionary systems.

#### 4.2.2. Protein Engineering Opportunities Arising from Complementary Modularity

Interactome formation by means of swapping and selecting for complementary modules has very important engineering applications that are well-known in some types of engineering (e.g., [157,158]) but which have barely been explored within receptor, transporter, and other types of protein evolution [159,160,161,162,163]. Understanding the modular basis of the transmembrane helices that determine membrane insertion [159,164], along with the conservation of complementarity-determining sequences that mediates their ability to form dimers and oligomers (e.g., this study), should permit the engineering of novel interactions between currently non-communicating receptor types. Similarly, the ability to swap in or swap out modules that determine allosteric control of receptors by molecules other than the preferred ligand should also permit novel control mechanisms to be explored or the decoupling of systems that normally engage in crosstalk. Altering receptor properties, such as their ability to sustain their native conformation without integrating into a lipid membrane, is possible by retaining the sequences that determine inter-helix contacts while modifying those that determine solvation properties ([165], and this volume). Novel receptors might be evolved analogously to the way enzymes are currently evolved for new functions, but by swapping of varied complementary modules instead of single amino acid substitutions. The homo- and heterocomplementary peptide subunits involved in complementary interactions within and between receptors could be used as the basis for creating self-assembling materials on the model of previous such materials [166,167,168,169,170,171,172,173]. Such experiments would also provide detailed tests of the model of receptor evolution proposed here.

Additional tests might involve attempts to recreate the origins of receptors and transporters along the lines suggested by our evolutionary model. Could a simple construct made up of one or two transmembrane modules and appropriate extracellular and intracellular linking modules with ligand binding affinity be sufficient to produce a functional receptor? If not, what would a minimal opioid or adrenergic receptor look like? These are questions that might best be addressed “bottom up” by trying to construct such a receptor module by module and also “top down” by successively eliminating modules, one by one. As noted above, some functional mu-OPR transcripts lack the first transmembrane module (helix 1) [113], and Qing, et al. [174] have already hinted at the possibilities inherent in the experimental approach in their discovery of severely truncated transcripts of the nfCCR5^QTY^ and nfCXCR4^QTY^ receptors that retain activity. Is the dispensability of TM1 also true for ADR and other aminergic receptors? What additional modules can be eliminated without losing all function? Could even simpler constructs such as an endorphin-like extracellular sequence attached to a single conserved transmembrane sequence with an intracellular “tail” comprise another endorphin-like sequence (Figure 20)? Or are two or more transmembrane helices required to provide some structural stability to the extracellular and intracellular loops and convey, through allosteric structural changes, the effects of extracellular binding of the ligand? Modular complementarity permits such questions to be posed in ways that can be addressed by conceptually simple experiments that do not require the usual random scanning of dozens or hundreds of individual amino acid positions.

Finally, swapping of modular, complementary regions permits us to begin to understand how evolution rapidly and efficiently evolves interactive sets of proteins so that they function as systems rather than randomly exploring the almost infinite possibilities of protein space and the selecting among the permutations. Selecting and re-using complementary modules in novel combinations assures interactivity and communication within protein systems. Understanding the principles of such complementary modular systems will mean that we will be able to harness these effects through molecular engineering to explore biological systems in new ways and more efficiently invent new biomedical applications.

## Figures and Tables

**Figure 1 life-11-01217-f001:**
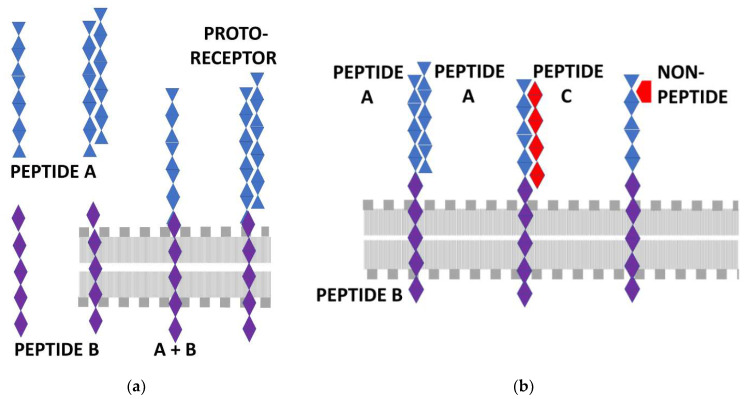
(**a**) Schematic representation of Dwyer’s original theory [59,60] of receptor–ligand co-evolution in which a peptide (A) self-aggregates (upper left) and one component of the dimer evolves into the receptor while the other evolves into the ligand. The receptor copy of peptide A oligomerizes with a peptide B that functions to anchor the protoreceptor A + B into the lipid membrane (represented by the gray dots and vertical lines), and this proto-receptor is able to bind the peptide A ligand copy. (**b**) Root-Bernstein’s extension of Dwyer’s theory to incorporate hetero-complementary peptides and non-peptides. Peptide A is homocomplementary to itself but hetero-complementary to a different peptide C; non-peptides, such as heterocyclic compounds or amines, may also be complementary to peptide A. Thus, both peptide C and complementary non-peptides may bind to the proto-receptor composed of the peptide A + B oligomer, and then selection pressures will act to evolve more specialized versions of the common receptor [60,63,64,65].

**Figure 2 life-11-01217-f002:**
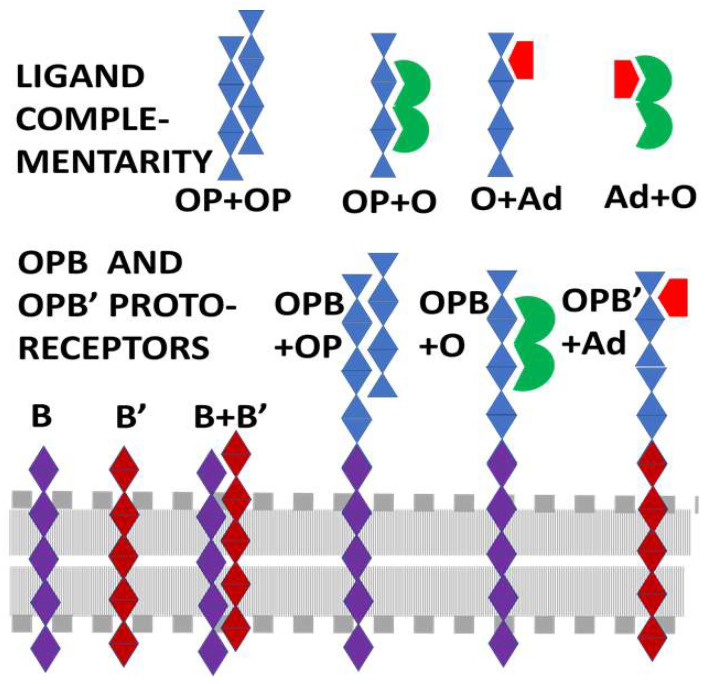
Graphical summary of predictions. Opioid peptides (OP) will self-aggregate (i.e., be homocomplementary) forming the basis for a receptor–ligand evolutionary pair. TOP ROW: Ligand complementarities will become the basis for receptor–ligand co-evolution. Since opioid drugs such as morphine (O) mimic opioid peptide (OP), the OP will become the basis for the O receptor. Adrenergic compounds (Ad) also bind to OP, so that OP will become the basis for the Ad receptor as well. Since Ad bind to OP, they will also bind to O. BOTTOM ROW: OP will become ligated to conserved peptides B and B’ selected evolutionarily for integration into the lipid membrane (represented by gray dots and lines). These membrane-spanning peptides will also be selected for their ability to aggregate (B + B’ heterocomplementarity). Proto-receptors will result from the ligation of OP sequences with B or B’ sequences. The extracellular OP sequence will be able to bind other OP, O, and Ad, providing the basis for the evolution of various opioid and adrenergic receptors from conserved modules. NOT SHOWN: Use of B and B’ membrane-spanning regions will make possible B + B’-mediated aggregation of various opioid and adrenergic receptors resulting in both homo- and heterodimerization.

**Figure 3 life-11-01217-f003:**
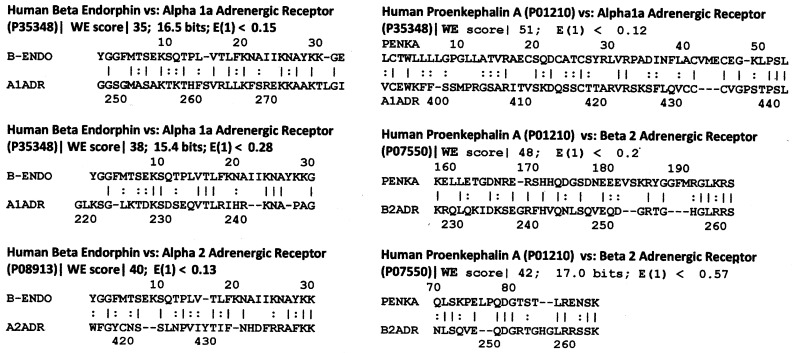
LALIGN homology search illustrating highly significant similarities between human beta endorphin (B-ENDO) and proenkephalin A (PENKA) with human adrenergic receptor sequences. UniProt accession numbers are shown in parentheses. Lines between sequences denote amino acid identities; double dots denote substitutions resulting in a structurally and chemically similar amino acid. LALIGN was accessed via the expasy.org website with the E value set to 10.0, opening gap penalty of −10.0, extending gap penalty of −2.0, 20 best matches to show, using the BLOSUM80 scoring matrix to maximize short, highly homologous sequence identification. Note that all Waterman–Eggert (WE) scores are 37 or greater and that all of the matches have a sequence in which there are five identities (a pair of similarities equals one identity) within a 10 amino acid stretch. This pair of characteristics makes these homologies statistically significant, as is discussed following Prediction 6 in the section on statistical tests.

**Figure 4 life-11-01217-f004:**
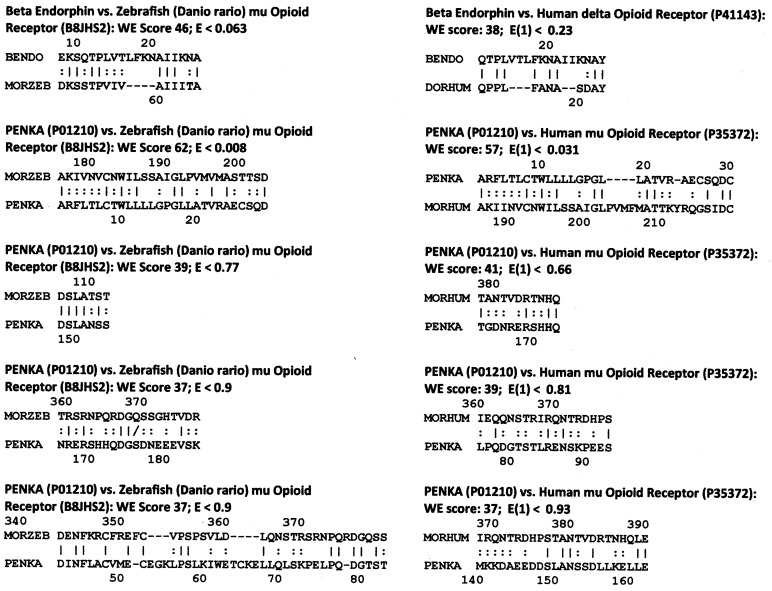
Selected LALIGN homology results comparing beta endorphin and proenkephalin A (PENKA) to the zebrafish (MORZEB) and human (MORHUM) opioid receptor sequences. UniProt accession numbers are shown in parentheses Lines between sequences denote amino acid identities; double dots denote substitutions resulting in a structurally and chemically similar amino acid. LALIGN was accessed via the expasy.org website with the E value set to 10.0, opening gap penalty of −10.0, extending gap penalty of −2.0, 20 best matches to show, using the BLOSUM80 scoring matrix to maximize short, highly homologous sequence identification. Note that all Waterman–Eggert (WE) scores are 37 or greater and that all of the matches have a sequence in which there are five identities (a pair of similarities equals one identity) within a 10 amino acid stretch. This pair of characteristics makes these homologies statistically significant, as is discussed following point 8 in the section on statistical tests.

**Figure 5 life-11-01217-f005:**
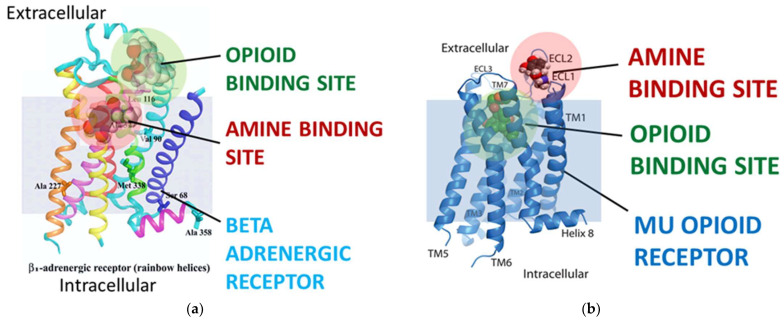
Ribbon models of the beta adrenergic receptor (**a**) and the mu opioid receptor (**b**) summarizing the main ligand binding sites and the most likely enhancer binding site for each receptor class according to the data reviewed in Predictions 4 and 5 (see text for details). The main point of this figure is to illustrate how the same regions of each receptor have evolved specificity for both classes of compounds, but that the specific sites of binding have been reversed. These models have been adapted from [41,77].

**Figure 6 life-11-01217-f006:**
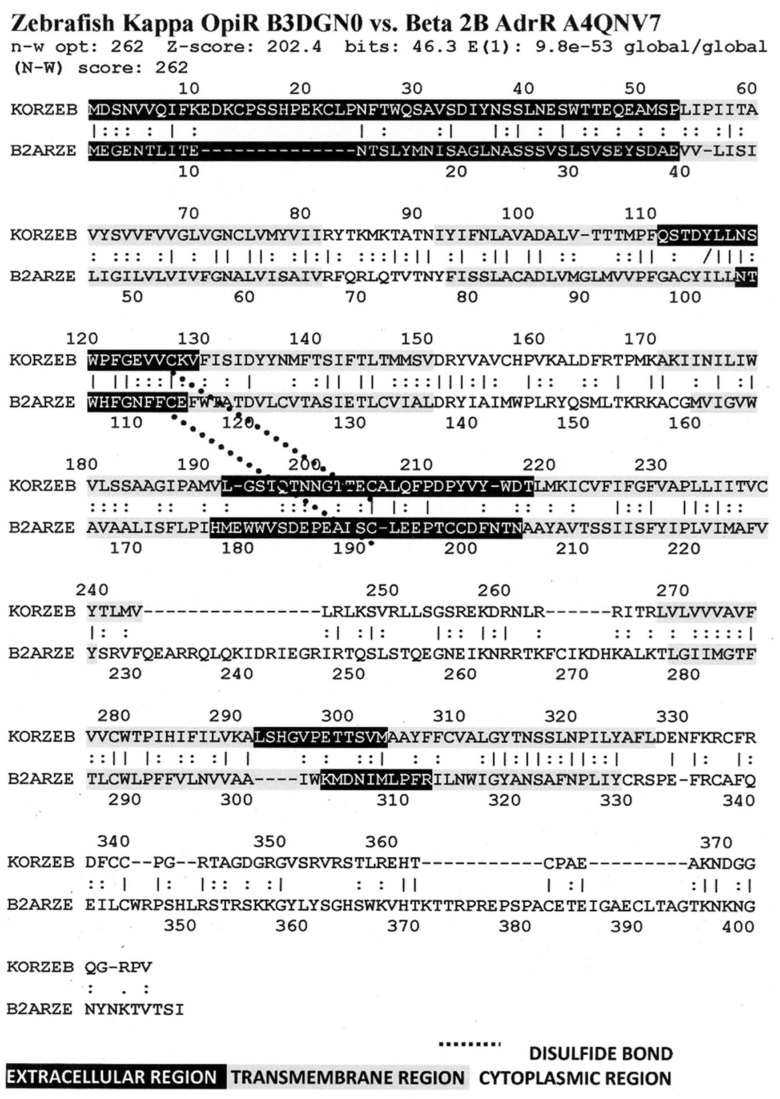
Homology between the zebrafish kappa opioid receptor (OpiR) and the zebrafish beta 2 adrenergic receptor (Beta 2 AdrR). The numbers following the receptors are the UniProt accession numbers. Black-outlined regions with white text represent extracellular (EC) regions; light grey regions with black text represent transmembrane (TM) regions; white regions with black text represent intracellular (IC) regions. Lines between the sequences represent amino acid identities while double dots represent conserved amino acid substitutions. The dotted lines connecting transmembrane regions represent disulfide bonds. LALIGN was accessed on the expasy.org website) setting the search algorithm to BLOSUM80 (optimized for locating short sequences of high homology), E value at 10.0, opening gap penalty of −12.0, extending gap penalty of −2.0, with a “global search without endpoint penalty” function. (Very similar, and sometime identical, results are obtained using the “local search” function with default settings, which suggests the robustness of the homologies reported below.) Figure 7, Figure 8 and Figure 9 used the same methodology. Comparisons were constrained to species for which defined opioid and adrenergic receptors have been sequenced, in addition to the availability of species-specific pro-opiomelonocortin and pro-enkephalin sequences.

**Figure 7 life-11-01217-f007:**
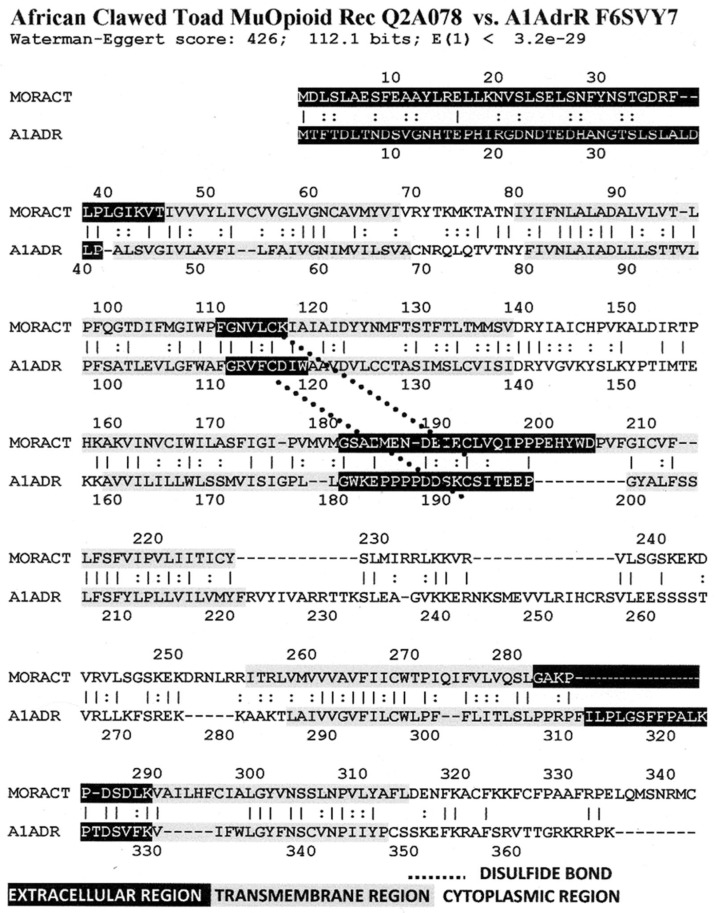
Homology between the African clawed toad mu opioid receptor (MuOpioid Rec) and the African clawed toad alpha 1A adrenergic receptor (A1AdrR). The numbers following the receptors are the UniProt accession numbers. Black-outlined regions with white text represent extracellular (EC) regions; light grey regions with black text represent transmembrane (TM) regions; white regions with black text represent intracellular (IC) regions. Lines between the sequences represent amino acid identities while double dots represent conserved amino acid substitutions. The dotted lines connecting transmembrane regions represent disulfide bonds. See Figure 6 for details of the methodology.

**Figure 8 life-11-01217-f008:**
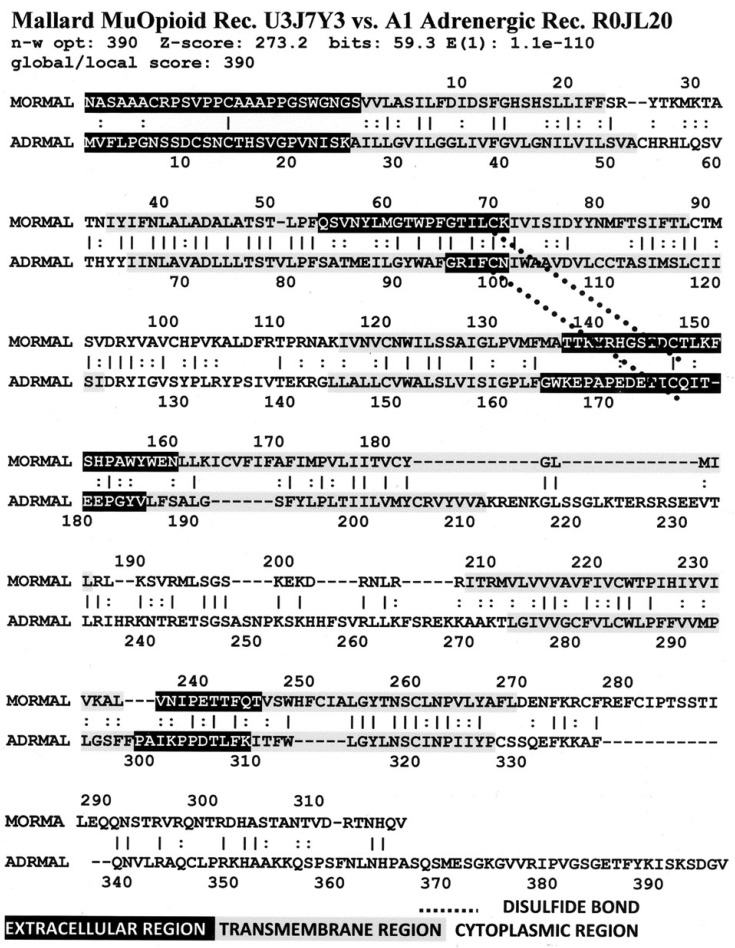
Homology between the mallard mu opioid receptor (MuOpioid Rec) and the mallard alpha 1A adrenergic receptor. The numbers following the receptors are the UniProt accession numbers. Black-outlined regions with white text represent extracellular (EC) regions; light grey regions with black text represent transmembrane (TM) regions; white regions with black text represent intracellular (IC) regions. Lines between the sequences represent amino acid identities while double dots represent conserved amino acid substitutions. The dotted lines connecting transmembrane regions represent disulfide bonds. See Figure 6 for details of the methodology.

**Figure 9 life-11-01217-f009:**
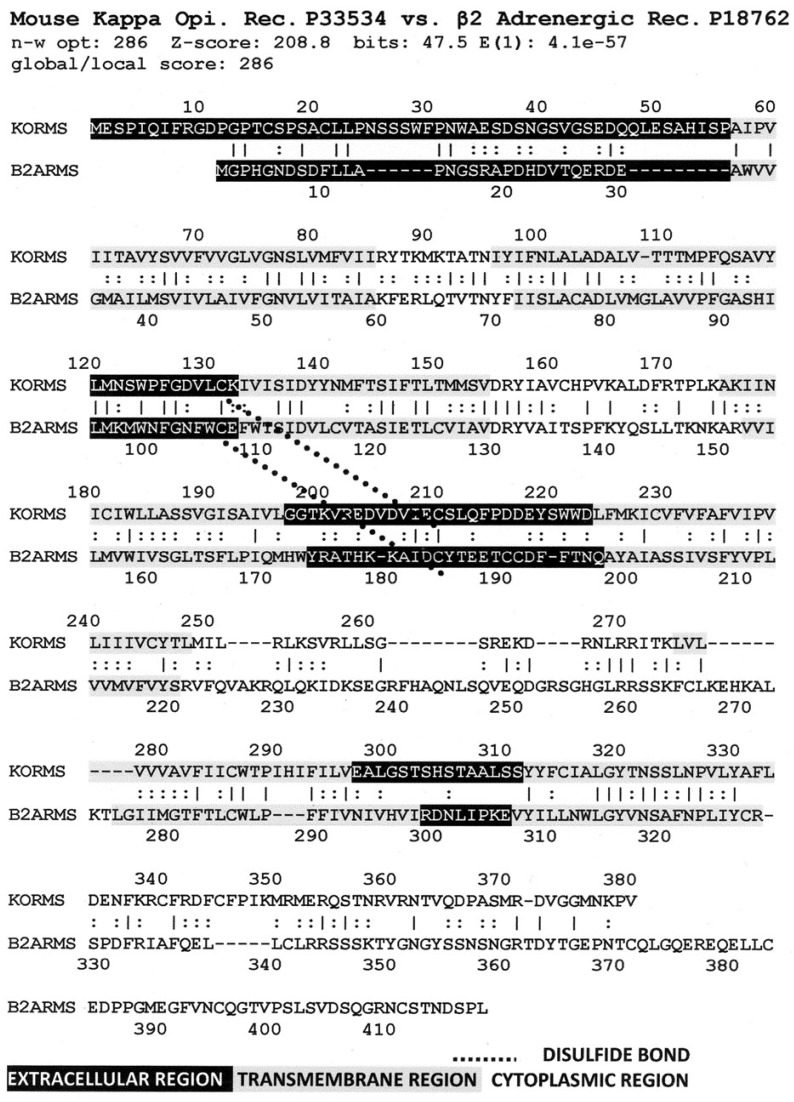
Homology between the mouse kappa opioid receptor (Kappa Opi Rec) and the mouse beta 2 adrenergic receptor. The numbers following the receptors are the UniProt accession numbers. Black-outlined regions with white text represent extracellular (EC) regions; light grey regions with black text represent transmembrane (TM) regions; white regions with black text represent intracellular (IC) regions. Lines between the sequences represent amino acid identities while double dots represent conserved amino acid substitutions. The dotted lines connecting transmembrane regions represent disulfide bonds. See Figure 6 for details of the methodology.

**Figure 10 life-11-01217-f010:**
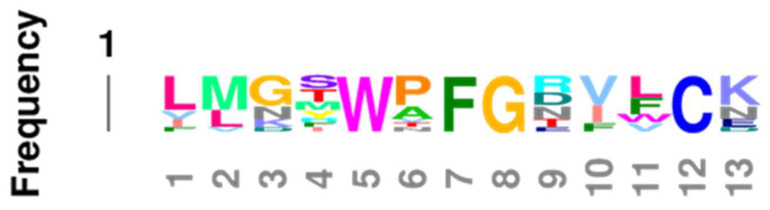
EC2 Logoplot for the seven opioid and seven adrenergic receptors used in this study. Logoplots were generated by aligning the sequences from each region of the fourteen receptor sequences used in this study (see text, particularly Figure 6, Figure 7, Figure 8 and Figure 9, Figure 16, Figure 17 and Figure 18) and then using kpLOGO [102]. The letters each represent an amino acid at that position and employ the standard one-letter code for amino acids. The size of the letter (from zero to one unit, left-hand axis) provides a measure of how frequently the amino acid is utilized at that position, with zero being never and one being always. Here the W at position 5, F at position 7, G at position 8, and C at position 12 are always present in all fourteen sequences used to generate the Logoplot.

**Figure 11 life-11-01217-f011:**
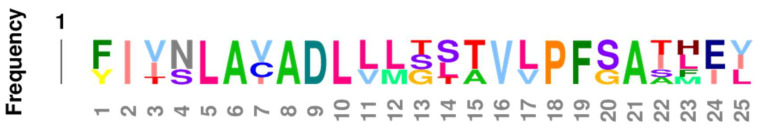
TM2 Logoplot for the seven opioid and seven adrenergic receptors used in this study. See Figure 10 for methods and explanation of axes.

**Figure 12 life-11-01217-f012:**
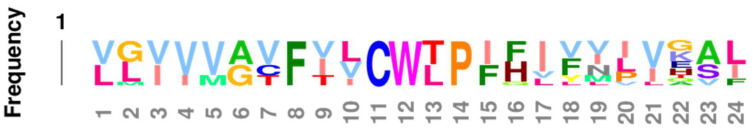
TM6 Logoplot for the seven opioid and seven adrenergic receptors used in this study. See Figure 10 for methods and explanation of axes.

**Figure 13 life-11-01217-f013:**
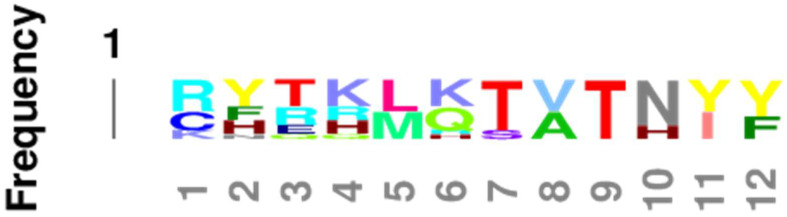
IC1 Logoplot for the seven opioid and seven adrenergic receptors used in this study. See Figure 10 for methods and explanation of axes.

**Figure 14 life-11-01217-f014:**
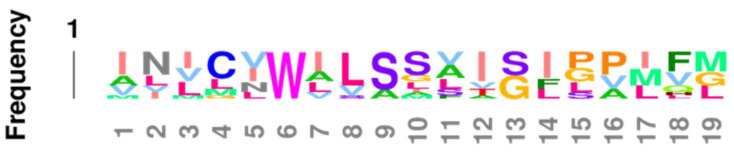
TM4 Logoplot for the seven opioid and seven adrenergic receptors used in this study. See Figure 10 for methods and explanation of axes.

**Figure 15 life-11-01217-f015:**
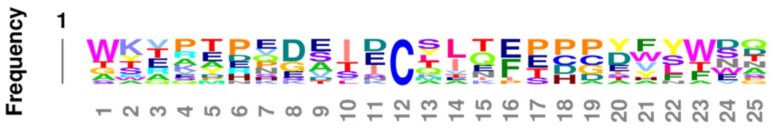
EC3 Logoplot for the seven opioid and seven adrenergic receptors used in this study. See Figure 10 for methods and explanation of axes.

**Figure 16 life-11-01217-f016:**
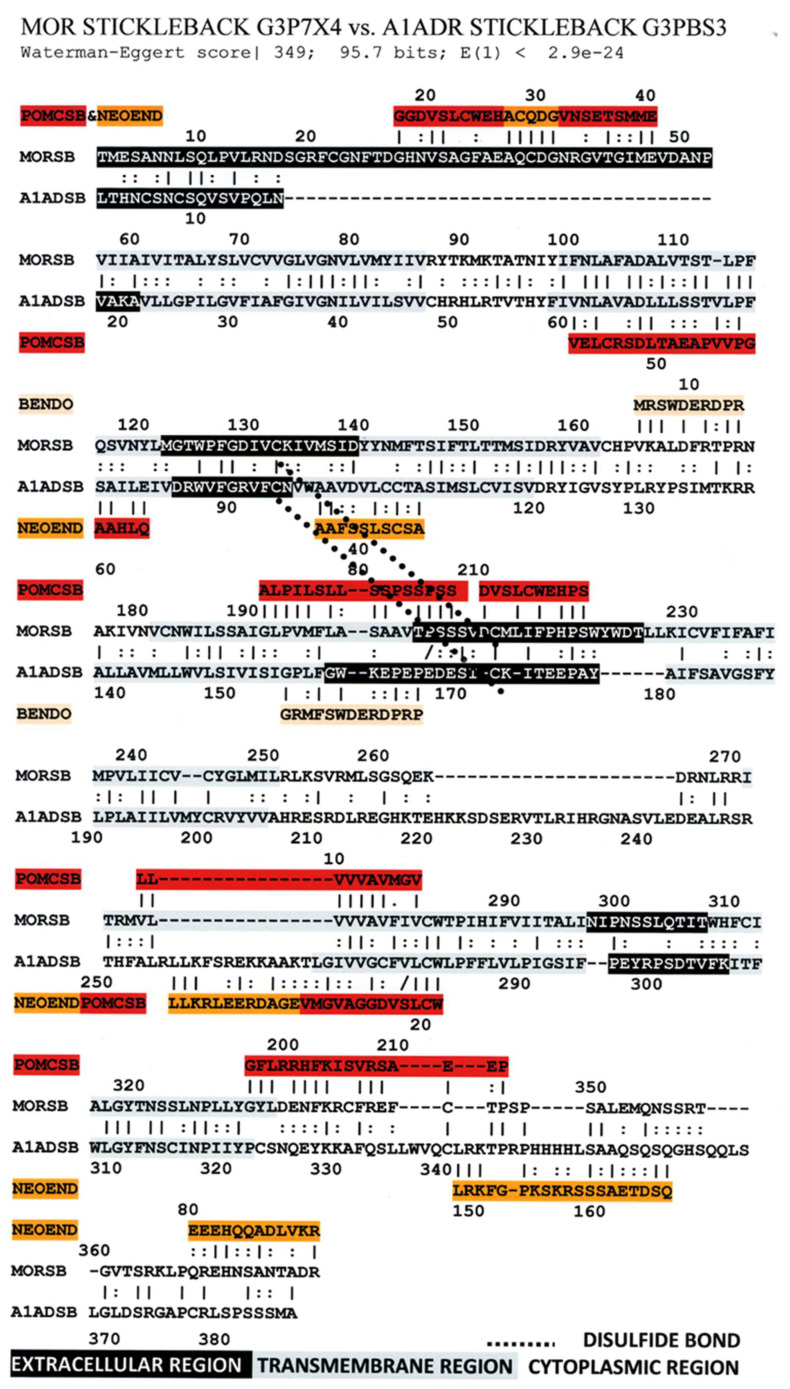
Homology between the stickleback mu opioid receptor (MORSB) and the stickleback alpha 1A adrenergic receptor (A1ADSB) supplemented with regions of opioid peptide homologies. The numbers following the receptors are the UniProt accession numbers. Sequences highlighted in color represent homologies to opioid precursor proteins: Pale orange = stickleback beta endorphin (BENDO); Orange = stickleback neoendorphin (NEOEND); Red = Stickleback pro-opiomelanocortin (POMCSB). Black-outlined regions with white text represent extracellular (EC) regions; light grey regions with black text represent transmembrane (TM) regions; white regions with black text represent intracellular (IC) regions. Lines between the sequences represent amino acid identities while double dots represent conserved amino acid substitutions. The dotted lines connecting transmembrane regions represent disulfide bonds. See Figure 6 for details of the methodology.

**Figure 17 life-11-01217-f017:**
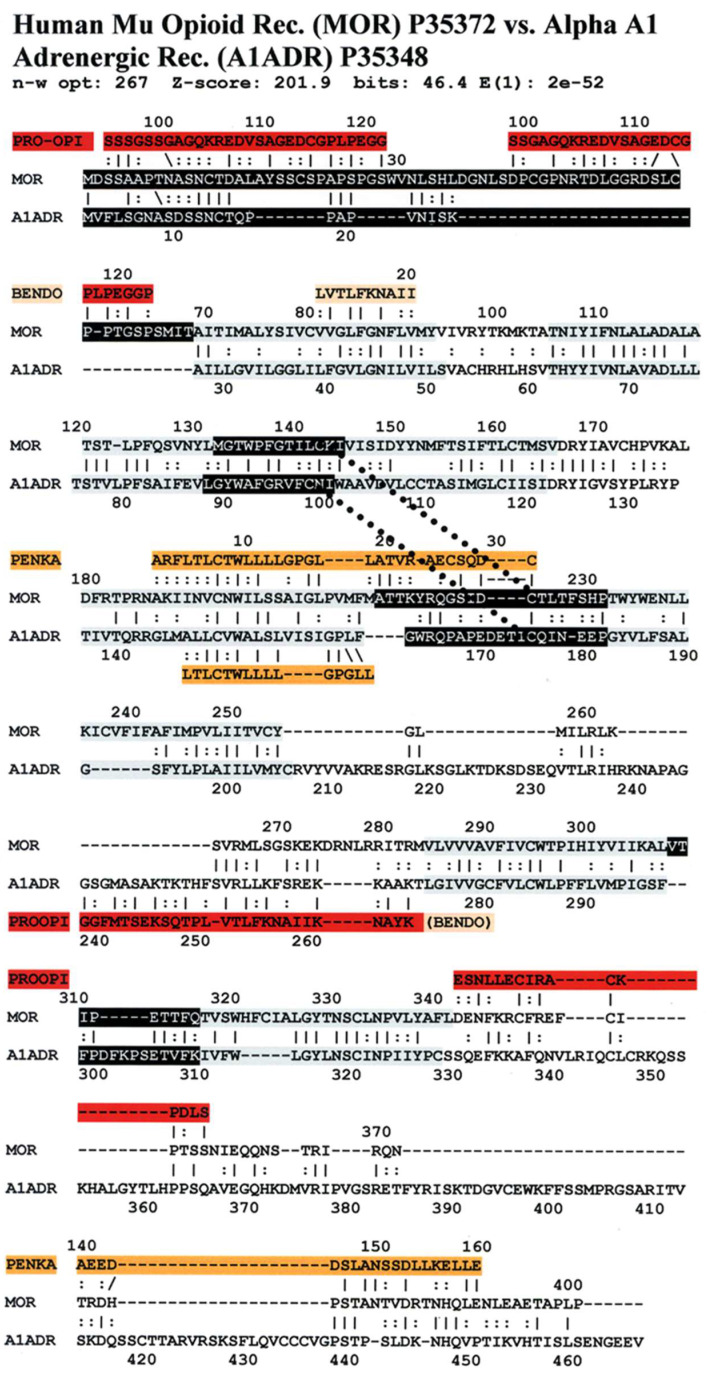
Homology between the human mu opioid receptor (MOR) and the human alpha 1A adrenergic receptor (A1ADR) supplemented with regions of opioid peptide homologies. The numbers following the receptors are the UniProt accession numbers. Sequences highlighted in color represent homologies to opioid precursor proteins: Pale orange = human beta endorphin (BENDO); Orange = human pro-enkephalin (PENKA); Red = human pro-opiomelanocortin (PROOPI). Black-outlined regions with white text represent extracellular (EC) regions; light grey regions with black text represent transmembrane (TM) regions; white regions with black text represent intracellular (IC) regions. Lines between the sequences represent amino acid identities while double dots represent conserved amino acid substitutions. The dotted lines connecting transmembrane regions represent disulfide bonds. See Figure 6 for details of the methodology.

**Figure 18 life-11-01217-f018:**
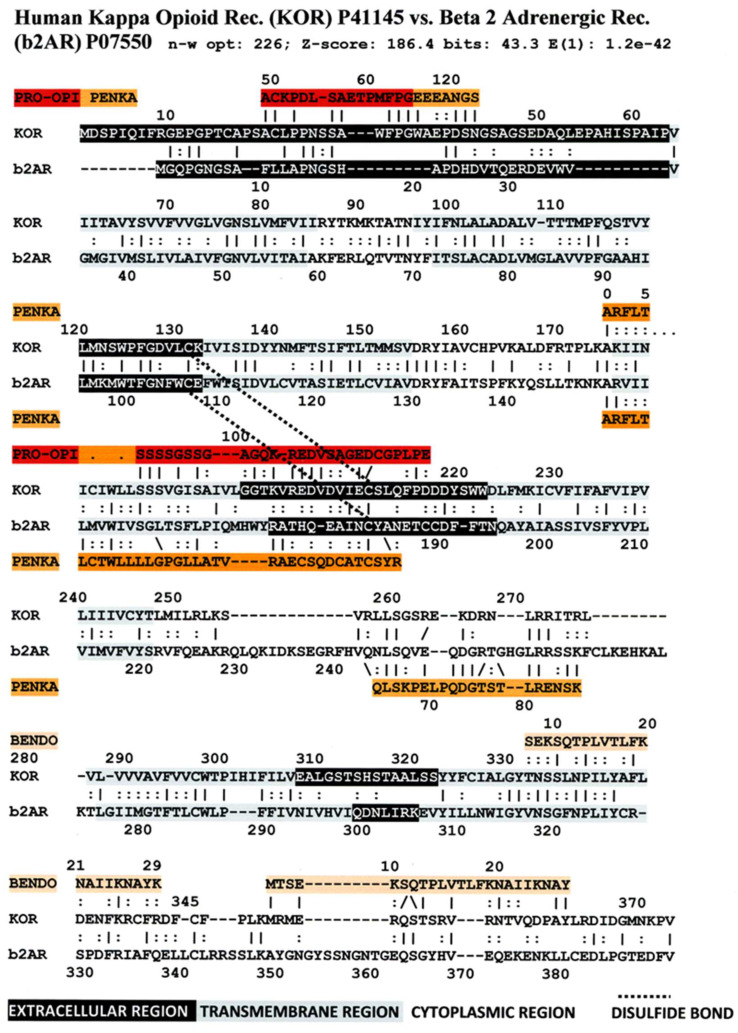
Homology between the human kappa opioid receptor (KOR) and the human beta 2 adrenergic receptor (b2AR) supplemented with regions of opioid peptide homologies. The numbers following the receptors are the UniProt accession numbers. Sequences highlighted in color represent homologies to opioid precursor proteins: Pale orange = human beta endorphin (BENDO); Orange = human pro-enkephalin (PENKA); Red = human pro-opiomelanocortin (PROOPI). Black-outlined regions with white text represent extracellular (EC) regions; light grey regions with black text represent transmembrane (TM) regions; white regions with black text represent intracellular (IC) regions. Lines between the sequences represent amino acid identities while double dots represent conserved amino acid substitutions. The dotted lines connecting transmembrane regions represent disulfide bonds. See Figure 6 for details of the methodology.

**Figure 19 life-11-01217-f019:**
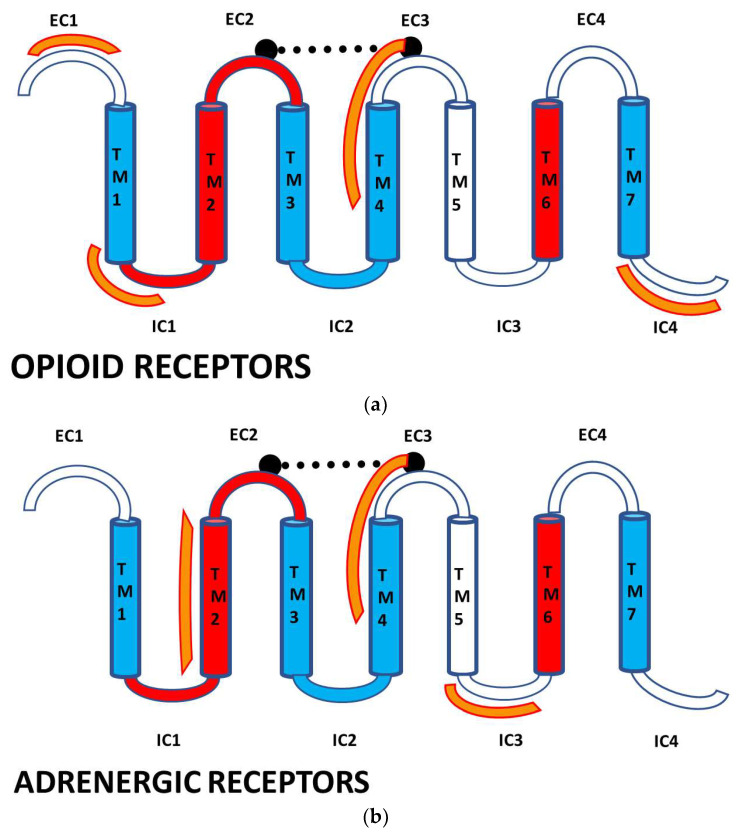
(**a**) Abstracted schematic figure summarizing the fundamental structural elements of opioid receptors abstracted from the detailed data found in Figure 6, Figure 7, Figure 8, Figure 9, Figure 10, Figure 11, Figure 12, Figure 13, Figure 14, Figure 15, Figure 16, Figure 17 and Figure 18 and Table 3. (**b**) Abstracted schematic figure summarizing the fundamental structural elements of adrenergic receptors abstracted from the detailed data found in Figure 6, Figure 7, Figure 8, Figure 9, Figure 10, Figure 11, Figure 12, Figure 13, Figure 14, Figure 15, Figure 16, Figure 17 and Figure 18 and Table 3. EC = extracellular region; TM = transmembrane region; IC = intracellular region; red regions are highly conserved (70% or greater) across both opioid and adrenergic receptors; blue regions are moderately conserved (60–69%) across both opioid and adrenergic receptors; white regions are poorly conserved (30–57%) across both opioid and adrenergic receptors; orange bars represent the main regions that are homologous to one or more opioid peptides; the black circles joined by dots represent the two conserved cysteine residues that participate in a disulfide bond that helps to form the ligand receptor pocket. Note that opioid peptide-like regions are not associated with the most conserved regions of the receptors but, with one exception in TM4-EC3, occur in different locations in the adrenergic versus the opioid receptors. Thus, while opioid peptide-like regions occur in both types of receptors, they occur in different arrangements suggesting modular swapping.

**Figure 20 life-11-01217-f020:**
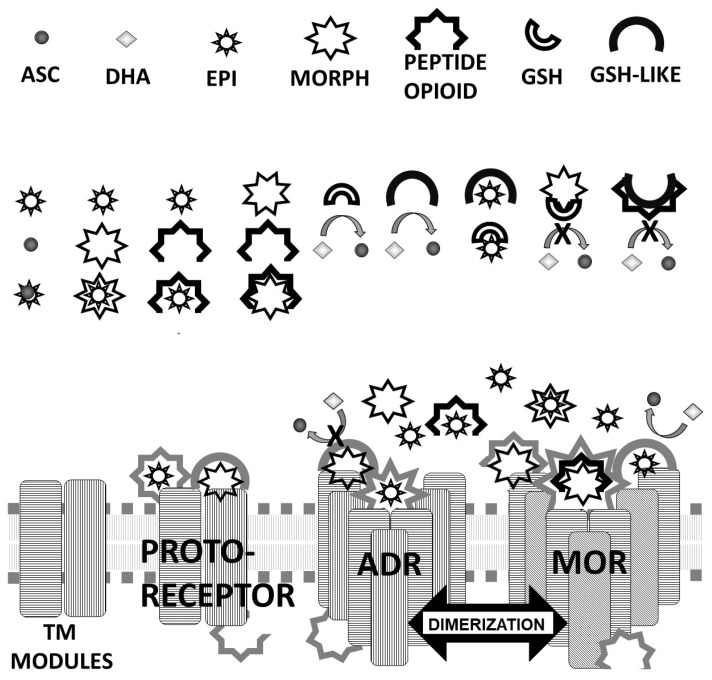
Schematic model of the main molecularly complementary interactions acting as selection criteria during the co-evolution of adrenergic and opioid receptors. The model assumes eight basic sets of compounds: ascorbic acid (ASC); dehydroascorbic acid (DHA); epinephrine and other adrenergic compounds such as norepinephrine, amphetamine, dopamine and their metabolites (EPI); peptide and non-peptide opioids (MORPH; PEPTIDE OPIOID); glutathione (GSH); glutathione-like polypeptides (GSH-LIKE); and peptides selected for their ability to insert stably into or across cell membranes (transmembrane sequences or TM MODULES). It is proposed that the complexity of opioid–adrenergic cross-talk results from this multitude of interactions that have been incorporated into the co-evolution of their receptors. Opioid peptide-like modules and GSH-like modules that have been integrated into the receptor sequences are presented in gray while ligands are presented in black. Note that there are several types of TM modules, some of which are complementary and permit homo- and hetero-dimerization. Note also the presence of opioid-peptide-like modules on the extracellular and intracellular regions of the opioid receptor that may function so that extracellular binding of opioids uncovers an opioid-like sequence on the intracellular side, thus conveying an opioid “message” across the membrane. The adrenergic receptor lacks this feature (see Figure 19). See text above for a detailed description of the chemical selection pressures driving integration of these molecularly complementary modules to be mixed and matched through ligation of small gene segments to create different classes of receptors that retain interactive functionality through conserved sets of homo- and hetero-complementarity. For simplicity, not every possible interaction discussed in the Results sections are summarized in this schematic so that, for example, wherever morphine binds, opioid peptides also bind and vice versa.

**Table 1 life-11-01217-t001:** Summary of binding studies of adrenergic compounds and other small molecules to opioid peptides and morphine. M-Enk = methionine enkephalin; Morph = morphine. Binding constants are Kd’s in micromoles. U. V. = ultraviolet spectroscopy; nm = nanometers (wavelength, adapted from [17,75,76,77] supplemented with previously unpublished endomorphin data using the same methods as described in [75,76,77]). * = Binding, or lack thereof, confirmed by NMR [75]; @ = Binding confirmed by capillary electrophoresis [76].

Kd (µM) U. V. @ 200 nm	M-Enk	Endomorphin	Morph
Epinephrine HCl	5.8 *	0.3	8.0 @ 0.5 *
Norepinephrine HCL	5.3 *	0.4	0.4 *
Dopamine	30 *	0.5	0.6 *
L-DOPA	70 *		>1000 *
Amphetamine	80 *		0.1 *
Propranolol	25		45
Salbutamol	30		0.3
Isoproterenol	40		0.1
Phenylephrine	30		0.13
Tyramine	12		50
Octopamine	80 *	0.35	3.2 *
Homovanillic Acid	80 *		>1000 *
Tyrosine	>1000 *		>1000 *
Phenylalanine	>1000 *		>1000 *
Serotonin	45 *	0.2	0.7 *
Melatonin	130	1.2	300
Histamine	>1000 *	0.17	>1000 *
Acetylcholine	80 *	>1000	>1000 *
Ascorbic Acid	600	>1000	>1000

**Table 2 life-11-01217-t002:** Summary of binding studies of adrenergic compounds, adrenergic precursors, and opioids to human mu opioid receptor (muOPR) peptides and human beta 2 adrenergic receptor (B2AR) peptides. Binding constants are Kd’s in micromoles. U. V. = ultraviolet spectroscopy; nm = nanometers (wavelength). Where more than one number is present, the binding curves were biphasic showing both high-affinity binding followed by a slash and then the low-affinity binding. EC = extracellular domain; TM = transmembrane domain; numbers represent the amino acid sequence according to the UniProt sequence. Mor = morphine; Nalox = naloxone; MENK = methionine enkephalin; Epi = epinephrine (adrenalin); NorEpi = norepinephrine (noradrenalin); Tyro = tyrosine; Phe = phenylalanine. Adapted from [16,41,44,77]. Opioid agonists and antagonists are in CAPS; adrenergic compounds have only the first letter capitalized; amino acid precursors of adrenergic compounds are lowercase italicized.

Kd (μM) @ 200 nm	MOR	NALOX	MENK	Epi	NorEpi	*tyro*	*phenyl*
Mu OPR 38–51, EC1	35	0.5/35	0.15/55	1.2/35	1.4/45	85	50
Mu OPR 111–122, TM2	50	0.5/38	0.33/80	1.3/40	1.3/40	700	>1000
Mu OPR 121–131, TM2	900	>1000	3.5/90	>1000	>1000	>1000	>1000
Mu OPR 132–143, EC2	35	0.5/42	0.4/70	1.4/35	1.4/40	60	80
Mu OPR 211–226, EC3	30	1.0/45	1.0/65	1.2/40	1.3/45	160	200
B2AR 97–103, EC2	1	6	130	120	600	>1000	>1000
B2AR 103–113, TM5	40	50				>1000	>1000
B2AR 105–108, EC2/TM5	30	30	30	35	35	50	55
B2AR 175–188, EC3	50	40	700	900	1000	>1000	>1000
B2AR 183–185, EC3	>1000	>1000	>1000	25	12	35	30

**Table 3 life-11-01217-t003:** Summary of percent conservation of amino acids in each receptor region of the fourteen adrenergic and opioid receptors used in this study for each species (SP); one each from zebrafish (ZEB), stickleback (STK), African clawed toad (TO), mallard (MAL), mouse (MS), human mu opioid receptor with human alpha A1 adrenergic receptor (HUM), and human kappa opioid receptor with human beta 2 adrenergic receptor (HUK). These species were chosen first to represent a range of taxonomic classes spanning vertebrate evolution and second, and more specifically, because the range of ligand and receptor sequences necessary to make the analyses was available, which is not the case for many species. EC = extracellular region (black regions in Figure 6, Figure 7, Figure 8 and Figure 9); TM = transmembrane region (grey regions in Figure 6, Figure 7, Figure 8 and Figure 9); IC = intracellular region (white regions in Figure 6, Figure 7, Figure 8 and Figure 9). Fractions in each box represent the number of conserved amino acids (identical or conserved substitutions)—the first number in each box—as a function of the total length of the receptor region—the second number in each box: conserved/total amino acids. The average percent of such conserved amino acids (Avg. Cons.) for all fourteen receptors is given in the bottom row.

SP	EC 1	EC 2	EC 3	EC 4	TM 1	TM 2	TM 3	TM 4	TM 5	TM 6	TM 7	IC 1	IC 2	IC 3	IC 4
ZEB	21/53	14/19	13/26	6/12	20/29	13/19	13/21	12/21	15/25	19/20	15/23	9/10	13/20	14/51	32/79
STK	13/52	11/18	8/20	4/11	18/29	21/25	17/22	14/20	13/25	22/28	14/22	9/12	8/14	11/51	32/64
TO	15/45	6/8	8/24	7/19	17/25	21/31	13/22	14/24	15/23	23/28	12/25	6/11	11/19	22/60	6/42
MAL	3/26	12/18	8/24	8/11	15/27	15/18	13/23	10/21	15/26	19/28	14/24	7/11	16/27	19/62	17/68
MS	17/56	9/13	12/26	3/15	19/29	17/25	14/22	16/23	14/25	17/26	15/23	10/11	14/21	21/54	25/90
HUM	19/68	10/13	12/19	7/12	14/24	20/28	12/21	13/27	14/28	16/24	14/23	4/11	10/14	18/67	41/137
HUK	18/63	9/13	16/26	4/15	17/26	16/24	14/22	15/24	14/25	19/24	15/23	9/11	13/21	18/54	21/64
Avg. Cons.	29%	70%	47%	41%	64%	72%	63%	60%	57%	76%	61%	70%	63%	31%	32%

**Table 4 life-11-01217-t004:** Human G-protein-coupled receptors (GPCR) identified by a BLAST search utilizing the Logoplot consensus sequences for the second extracellular receptor loop (EC2), second transmembrane helix (TM2), and sixth transmembrane helix (TM6) using the following criteria: BLAST searched against the “Homo sapiens” protein data, set to BLOSUM80, E = 10, no gaps, 1000 best scoring and best alignments to show. Almost identical results were found when the consensus sequences were searched against the vertebrate database as a whole (3000 best scoring and alignments to show). Bolded GPCR were homologous to all three of the conserved regions tested.

HUMAN	Homologous to:	Homologous to:	Homologous to:
Location	Extracellular 2 loop	Transmembrane helix 2	Transmembrane helix 6
Sequence	LMGSWPFGRVLCK	FIVNLAVADLLLTSTVLPFSA	VVAVFVLCWTPIFI
Receptors		Adenosine	
	**Adrenergic, Alpha**	**Adrenergic, Alpha**	**Adrenergic, Alpha**
	**Adrenergic, Beta**	**Adrenergic, Beta**	**Adrenergic, Beta**
	Angiotensin II	Angiotensin II	
		Cannabinoid	
	C3a anaphylatoxin		C3a anaphylatoxin
		C-X-C chemokine	C-X-C chemokine
		Cholecystokinin	Cholecystokinin
		Dopamine	Dopamine
	fMet-Leu-Phe		
	**Histamine**	**Histamine**	**Histamine**
		Melanocortin	Melanocortin
		Melatonin	
		Neuropeptide FF	
	Neuropeptide Y	Neuropeptide Y	
	Neuropeptides B/W	Neuropeptides B/W	
	Nociceptin		Nociceptin
	**Opioid, Delta**	**Opioid, Delta**	**Opioid, Delta**
	**Opioid, Kappa**	**Opioid, Kappa**	**Opioid, Kappa**
	**Opioid, Mu**	**Opioid, Mu**	**Opioid, Mu**
	Orexin	Orexin	
	P2Y purinoceptor		
	Relaxin		Relaxin
	**Serotonin**	**Serotonin**	**Serotonin**
	**Somatostatin**	**Somatostatin**	**Somatostatin**
		Thyrotropin-releasing hormone	
		Urotensin II	
			Vasopressin

## Data Availability

All data utilized in this study are provided in it or in the Appendix A and Appendix B.

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
