# Peer review of "Co-Evolution of Opioid and Adrenergic Ligands and Receptors: Shared, Complementary Modules Explain Evolution of Functional Interactions and Suggest Novel Engineering Possibilities"

_life, 2021, doi:10.3390/life11111217_

Round 1

Reviewer 1 Report

The authors propose that the opioid and adrenergic receptors evolved via the swapping of common modules as directed by molecular complementarity of the signalling molecules and their binding sites. This proposal is part of a more general hypothesis that extends to the evolution of other classes of receptors and to the evolution of the interactome itself. Evidence is provided in the form of sequence comparisons and experimental predictions are made. The paper is original, convincing and important and has major implications for the operation of receptors in modern cells. It is very stimulating and opens the door to other investigations such as the role of possible post-translational modifications associated with the modules.

Although the paper is in general clear, it would be easier to follow their arguments right from the start if they were to illustrate exactly is meant by ‘complementarity’ in the context of the opioid and adrenergic systems. Consider, for example, a signalling peptide, A, that binds to itself (A-A); consider too that in such binding to itself, either the first half of one A binds to the first half of the other A to give a homodimer or the first half of one A binds to the second half of the other A to give, perhaps, a filament. Now consider that signalling peptide A binds to sites on its receptor which can have the same sequence (so labelled ‘A’) or a different sequence, Ac (for A complement) to give either A-A or A-Ac. Which are we dealing with here in the case of the opioid receptor? Now consider a second signalling peptide, B, which also makes a homodimer, B-B, and interacts with its receptor sites to give either B-B or B-Bc. If B corresponds to an adrenergic ligand, such that there is A-B, which of the above possibilities are being investigated in this paper? An initial figure would help.

Specific comments

48 ‘in a manner that is not blocked by naloxone’ Perhaps ‘in a manner that is not blocked by the opioid antagonist, naloxone (which binds to the opioid receptor) ...’

50 ‘other research, however,’ Why ‘however’?

111 ‘volume’. There are many very different papers in a volume of Life so perhaps ‘special issue’?

134 ‘insulin and glucagon (which bind to each other with high affinity),’ and which have very different sequences?

161 ‘will’. Perhaps ‘should’

170-173 And reciprocally for the adrenergic regions of opioid receptors and adrenergic receptors? At this point, I am unsure whether these regions are the same or not. More generally, are the sequences of these regions similar (as in Dwyer) or different? A figure here that showed the predictions would be helpful.

204 “* = verified by nuclear magnetic resonance studies. ^ = verified by capillary electrophoresis studies” This is in duplicate.

Table 1 first line using ‘^’ like this risks confusion (10^3 = 10 to the power 3)

251 ‘significant homologies do exist between beta endorphin and PENKA with the mu and delta opioid receptors’ Try ‘the beta endorphin and PENKA peptides have significant homologies to on another and to the mu and delta opioid receptors’

253 ‘these homologies are conserved from fish through human beings (FIGURE 2).’ Why did they choose zebrafish?

258 ‘opioids often interact transiently with these extracellular loops prior to being drawn into their high-affinity binding site within the receptor’ Are these sites considered intracellular? They might clarify this here as it is not easy to understand why ligands that are central to extracellular signalling should have intracellular binding sites.

Figure 2. The order of Bendo compared to Dorhum and of Penka to Morhum should be reversed to be consistent with the other eight comparisons.

276 ‘Some opioid-like regions of opioid receptors should bind adrenergic compounds acting as allosteric modifiers of opioid receptor function.’ Do opioids also bind to the very same regions to act as allosteric modifiers?

313 ‘Epi = epinephrine (noradrenalin)’ Adrenalin?

Table 2 To help the uninitiated, use italics to distinguish between opioids and adrenergic compounds in the headings.

317 ‘Some of the opioid-like regions of adrenergic receptors should bind opioids’ As in line 276, are these the same regions that bind adrenergic compounds?

335 ‘The presence of a cysteine residue appears to be of particular importance’ My first reaction to this was ‘so what?’. In the light of the cysteine bridges in the following figures, maybe add ‘in structuring the protein (see below)’ (if this is the case).

Figure 3. I didn’t find this figure very helpful. Where are the intracellular regions referred to later? Where is TM2? Are all the putative binding sites shown? A simple line drawing for both receptors could be clearer.

‘A number of features are shared by all seven of the homology pairs we studied’ Try ‘by all seven pairs of the homology pairs of the opioid and adrenergic receptors we studied’.

Table 2. Is there any particular reason for the organisms that were chosen?

443 (and 447) ‘of the seven species used to generate TABLE 3’ Actually, six species but seven comparisons.

461 Figures

465 ‘F at position 7’

483 Keep ‘ligand’ as one word if possible

486 ‘has evolved in this region’ Which region – EC2 or EC3 or both?

491 Perhaps ‘the highest scoring consensus sequences’?

501 ‘GPCR’ in full

Table 4 why are some compounds in bold?

512 ‘several of the highly conserved regions correspond to sequences that are homologous to the opioid precursor proteins’ Try ‘several of the sequences of the highly conserved regions are homologous to the sequences of the opioid precursor proteins ... as demonstrated ...’

515 ‘Figures 14 through 16’ There is no Figure 14 so either the figure is missing or, more likely, there is a numbering problem and the figures are mislabelled as 15 to 17. (see also 550, 563 etc.)

553 ‘the results of the Logoplots’ it would help to add ‘for EC2, EC3, TM2, TM4, TM6 and IC1’

563 ‘FIGURES 14-16 clearly emphasize the point made by FIGURES 1-3 that these opioid peptide-receptor similarities occur at multiple sites in the receptors and that these similar regions occur both in the extracellular (including the first extracellular sequence, EC1, as well as EC3) and the fourth intracellular region (IC4) of the opioid receptors but only in one extracellular loop (EC3) and the third intracellular region (IC3) of the adrenergic receptors. Thus, these modules appear to have been rearranged.’ This is important but no schematic figure brings it all together and actually shows these similarities within the regions labelled EC1 etc.

582 ‘all of the transmembrane regions are conserved ... This conservation is also apparent in FIGURES 4-7 and 14-16 and is independent of the opioid-peptide modules.’ I am puzzled because in Figure 17, the region 10-20 of the opioid peptide BENDO appears to have an homology to the transmembrane regions 320-340 of KOR and b2AR (pale grey in the figure).

593-643 ‘The question now becomes whether these highly conserved TM regions are particularly important determinants of receptor dimerization etc.’ Receptors are often located in membrane rafts with a particular composition of lipids and proteins. Perhaps the question they pose requires lipid affinities to be taken into account.

740 ‘obtained by us and by Wolf ...’

827 ‘Some intracellular sequences may have been similar to the extracellular binding sequence so that the “message” conveyed by binding to the extracellular module was also replicated internally. This “internal image” would explain the presence of intracellular opioid peptide-like sequences corresponding to the extracellular opioid peptide-like sequences in both the adrenergic and opioid receptor classes.’ Presumably, the early proto-receptors contained an extracellular binding sequence with a general affinity for signalling molecules that duplication/rearrangement then put inside the cell.

Figure 18 This is a complicated figure that would be more helpful if it were restructured. For example, part (a) would be the seven compounds, part (b) the four sets of interactions (ASC with EPI etc.), part (c) the two reactions (GSH catalysing DHA into ASC etc.), part (d) the inhibition of the reactions in (c) by the binding of MORPH and PEPTDE OPIOID to the GSH and GSH-like compounds), and part (e) the rest of the figure. In this ‘rest of the figure’, I don’t know whether the peptide opioids shown attached to the intracellular sides of the proto-receptor, ADR and MOR are supposed to represent the opioid themselves or the opioid-like sites on these proteins; the GSH-like compound on ADR is shown catalysing its reaction though it is bound to MORPH which is supposed to antagonise this catalysis (line 795)

888 ‘One of the greatest mysteries that the mutation accumulation model leaves unresolved is why the random exploration of permutations does not lead to increasing loss of function over time and ever less integration within living systems when, instead, what is clearly observed across evolutionary time is an ever-increasing tendency toward systems integration exemplified in very stable interactomes. Selection for complementary modules can answer this conundrum’ and 902 ‘Interactome formation by means of swapping and selection for complementary modules produces, instead, a branched network of connections among evolutionary systems’ This is a powerful and persuasive idea. Perhaps supported by the work of Stuart Kauffman?

Author Response

REVIEWER 1:

LET US BEGIN BY THANKING THIS REVIEWER FOR TAKING A GREAT DEAL OF TIME WITH OUR MANUSCRIPT AND PROVIDING TRULY USEFUL AND INSIGHTFUL COMMENTS AND SUGGESTIONS. THIS IS PEER REVIEW AT ITS BEST AND OUR PAPER WILL BENEFIT ENORMOUSLY FROM THE CARE AND ATTENTION BESTOWED BY THIS REVIEWER.

The authors propose that the opioid and adrenergic receptors evolved via the swapping of common modules as directed by molecular complementarity of the signalling molecules and their binding sites. This proposal is part of a more general hypothesis that extends to the evolution of other classes of receptors and to the evolution of the interactome itself. Evidence is provided in the form of sequence comparisons and experimental predictions are made. The paper is original, convincing and important and has major implications for the operation of receptors in modern cells. It is very stimulating and opens the door to other investigations such as the role of possible post-translational modifications associated with the modules.

Although the paper is in general clear, it would be easier to follow their arguments right from the start if they were to illustrate exactly is meant by ‘complementarity’ in the context of the opioid and adrenergic systems. Consider, for example, a signalling peptide, A, that binds to itself (A-A); consider too that in such binding to itself, either the first half of one A binds to the first half of the other A to give a homodimer or the first half of one A binds to the second half of the other A to give, perhaps, a filament. Now consider that signalling peptide A binds to sites on its receptor which can have the same sequence (so labelled ‘A’) or a different sequence, Ac (for A complement) to give either A-A or A-Ac. Which are we dealing with here in the case of the opioid receptor? Now consider a second signalling peptide, B, which also makes a homodimer, B-B, and interacts with its receptor sites to give either B-B or B-Bc. If B corresponds to an adrenergic ligand, such that there is A-B, which of the above possibilities are being investigated in this paper? An initial figure would help.

EXCELLENT SUGGESTION: WE HAVE ADDED SUCH A FIGURE (NEW FIGURE 1) WITH A FIGURE CAPTION LAYING THIS OUT MORE CLEARLY.

 Specific comments

48 ‘in a manner that is not blocked by naloxone’ Perhaps ‘in a manner that is not blocked by the opioid antagonist, naloxone (which binds to the opioid receptor) ...’

ADDED

50 ‘other research, however,’ Why ‘however’?

‘HOWEVER’ DELETED

111 ‘volume’. There are many very different papers in a volume of Life so perhaps ‘special issue’?

SPECIAL ISSUE SUBSTITUTED

134 ‘insulin and glucagon (which bind to each other with high affinity),’ and which have very different sequences?

ADDED

161 ‘will’. Perhaps ‘should’

CORRECTED TO SHOULD

170-173 And reciprocally for the adrenergic regions of opioid receptors and adrenergic receptors? At this point, I am unsure whether these regions are the same or not. More generally, are the sequences of these regions similar (as in Dwyer) or different? A figure here that showed the predictions would be helpful.

FIGURE ADDED WITH EXPLANATORY CAPTION (NEW FIGURE 2).

204 “* = verified by nuclear magnetic resonance studies. ^ = verified by capillary electrophoresis studies” This is in duplicate.

DUPLICATION DELETED

Table 1 first line using ‘^’ like this risks confusion (10^3 = 10 to the power 3)

“@” SUBSTITUTED FOR ^

251 ‘significant homologies do exist between beta endorphin and PENKA with the mu and delta opioid receptors’ Try ‘the beta endorphin and PENKA peptides have significant homologies to one another and to the mu and delta opioid receptors’

SUBSTITUTION MADE! THANKS FOR ADDING CLARITY

253 ‘these homologies are conserved from fish through human beings (FIGURE 2).’ Why did they choose zebrafish?

ADDED FOR CLARIFICATION: “a fact that is explored in additional vertebrate species below. (Zebrafish are used as an example here because of they represent that most evolutionarily divergent species from Homo sapiens for which the range of opioid and adrenergic receptor and ligand sequences were available).”

258 ‘opioids often interact transiently with these extracellular loops prior to being drawn into their high-affinity binding site within the receptor’ Are these sites considered intracellular? They might clarify this here as it is not easy to understand why ligands that are central to extracellular signalling should have intracellular binding sites.

SORRY FOR THE CONFUSION. THE MAIN BINDING SITES FOR OPIOIDS ARE WITHIN A POCKET FORMED BY THE TRANSMEMBRANE REGIONS OF THE RECEPTORS. THIS POINT IS NOW MADE IN THE TEXT AND ILLUSTRATED IN FIGURE 3.

Figure 2. The order of Bendo compared to Dorhum and of Penka to Morhum should be reversed to be consistent with the other eight comparisons.

DONE

276 ‘Some opioid-like regions of opioid receptors should bind adrenergic compounds acting as allosteric modifiers of opioid receptor function.’ Do opioids also bind to the very same regions to act as allosteric modifiers?

AS NOTED ABOVE AT LINE 258, OPIOIDS DO BIND TRANSIENTLY WITH THESE SAME REGIONS BUT, AS FAR AS I KNOW, AN OPIOID CANNOT ACT AS AN ALLOSTERIC MODIFIER OF ITS OWN ACTIVITY.

313 ‘Epi = epinephrine (noradrenalin)’ Adrenalin?

CORRECTED

Table 2 To help the uninitiated, use italics to distinguish between opioids and adrenergic compounds in the headings.

ADDED: “Opioids agonists and antagonists are in CAPS; adrenergic compounds have only the first letter capitalized; and amino acid precursors of adrenergic compounds are lowercase italicized.”

317 ‘Some of the opioid-like regions of adrenergic receptors should bind opioids’ As in line 276, are these the same regions that bind adrenergic compounds?

THEY MAY OR MAY NOT (EVOLUTION HAS ADAPTED SOME REGIONS FOR BETTER SPECIFICITY OF BINDING TO ONE CLASS OF COMPOUNDS OVER THE OTHER). THIS CLARIFICATION HAS BEEN ADDED TO THE TEXT.

335 ‘The presence of a cysteine residue appears to be of particular importance’ My first reaction to this was ‘so what?’. In the light of the cysteine bridges in the following figures, maybe add ‘in structuring the protein (see below)’ (if this is the case).

REWORDED AS: “]. The presence of a cysteine residue appears to be of particular importance for the for-mation of a disulfide bond that keeps the receptor in its highest activity conformation.”

Figure 3. I didn’t find this figure very helpful. Where are the intracellular regions referred to later? Where is TM2? Are all the putative binding sites shown? A simple line drawing for both receptors could be clearer.

THE PURPOSE OF THIS FIGURE IS SIMPLY TO PROVIDE A SENSE OF WHERE THE MAIN BINDING SITES ARE FOR OPIOIDS AND ADRENERGICS IN EACH CLASS OF RECEPTOR (TO SOME EXTENT ANSWERING THE PROBLEM RAISED AT LINE 258 ABOVE. THERE IS NO INTENT TO PROVIDE DETAILS OF THE STRUCTURES HERE. THIS IS NOW CLARIFIED IN THE FIGURE CAPTION.

‘A number of features are shared by all seven of the homology pairs we studied’ Try ‘by all seven pairs of the homology pairs of the opioid and adrenergic receptors we studied’.

REWORDED AS SUGGESTED

Table 2. Is there any particular reason for the organisms that were chosen?

TABLE 3 FIGURE CAPTION? ADDED: “These species were chosen first to represent a range of taxonomic classes spanning vertebrate evolution and second, and more specifically, because the range of ligand and receptor sequences necessary to make the analyses was available, which is not the case for many species.”

443 (and 447) ‘of the seven species used to generate TABLE 3’ Actually, six species but seven comparisons.

CORRECTED

461 Figures

CORRECTED

465 ‘F at position 7’

CORRECTED

483 Keep ‘ligand’ as one word if possible

I CAN’T MODIFY THIS IN MY PROGRAM BUT I DON’T THINK IT WILL REMAIN AFTER FORMATING BY THE JOURNAL EDITOR…

486 ‘has evolved in this region’ Which region – EC2 or EC3 or both?

CORRECTED TO: “in these two regions”

491 Perhaps ‘the highest scoring consensus sequences’?

ADDED

501 ‘GPCR’ in full

Table 4 why are some compounds in bold?

ADDED: “Bolded GPCR were homologous to all three of the conserved regions tested.”

512 ‘several of the highly conserved regions correspond to sequences that are homologous to the opioid precursor proteins’ Try ‘several of the sequences of the highly conserved regions are homologous to the sequences of the opioid precursor proteins ... as demonstrated ...’

CORRECTED

515 ‘Figures 14 through 16’ There is no Figure 14 so either the figure is missing or, more likely, there is a numbering problem and the figures are mislabelled as 15 to 17. (see also 550, 563 etc.)

CORRECTED (FIGURE CAPTIONS WERE INCORRECTLY LABELLED)

553 ‘the results of the Logoplots’ it would help to add ‘for EC2, EC3, TM2, TM4, TM6 and IC1’

ADDED

563 ‘FIGURES 14-16 clearly emphasize the point made by FIGURES 1-3 that these opioid peptide-receptor similarities occur at multiple sites in the receptors and that these similar regions occur both in the extracellular (including the first extracellular sequence, EC1, as well as EC3) and the fourth intracellular region (IC4) of the opioid receptors but only in one extracellular loop (EC3) and the third intracellular region (IC3) of the adrenergic receptors. Thus, these modules appear to have been rearranged.’ This is important but no schematic figure brings it all together and actually shows these similarities within the regions labelled EC1 etc.

EXCELLENT POINT! WE HAVE ADDED A NEW FIGURE THAT SUMMARIZES THE KEY STRUCTURAL ELEMENTS TO HIGHLIGHT WHICH REGIONS ARE CONSERVED, WHICH ARE NOT, AND WHERE THE MAIN OPIOID-PEPTIDE-LIKE REGIONS ARE LOCATED.

582 ‘all of the transmembrane regions are conserved ... This conservation is also apparent in FIGURES 4-7 and 14-16 and is independent of the opioid-peptide modules.’ I am puzzled because in Figure 17, the region 10-20 of the opioid peptide BENDO appears to have an homology to the transmembrane regions 320-340 of KOR and b2AR (pale grey in the figure).

WELL, IT MATCHES KOR (ACTUALLY RATHER POORLY GIVEN THE LACK OF CONSERVED AMINO ACIDS) BUT IT DOES NOT MATCH THE b2AR (WHICH IS WHY IT DOESN’T APPEAR BESIDE THE b2AR SEQUENCE AS WELL). 

593-643 ‘The question now becomes whether these highly conserved TM regions are particularly important determinants of receptor dimerization etc.’ Receptors are often located in membrane rafts with a particular composition of lipids and proteins. Perhaps the question they pose requires lipid affinities to be taken into account.

A SENTENCE TO THIS EFFECT HAS NOW BEEN ADDED TO THE END OF THIS SECTION.

740 ‘obtained by us and by Wolf ...’

CORRECTED

827 ‘Some intracellular sequences may have been similar to the extracellular binding sequence so that the “message” conveyed by binding to the extracellular module was also replicated internally. This “internal image” would explain the presence of intracellular opioid peptide-like sequences corresponding to the extracellular opioid peptide-like sequences in both the adrenergic and opioid receptor classes.’ Presumably, the early proto-receptors contained an extracellular binding sequence with a general affinity for signalling molecules that duplication/rearrangement then put inside the cell.

A SENTENCE TO THIS EFFECT HAS BEEN ADDED.

Figure 18 This is a complicated figure that would be more helpful if it were restructured. For example, part (a) would be the seven compounds, part (b) the four sets of interactions (ASC with EPI etc.), part (c) the two reactions (GSH catalysing DHA into ASC etc.), part (d) the inhibition of the reactions in (c) by the binding of MORPH and PEPTDE OPIOID to the GSH and GSH-like compounds), and part (e) the rest of the figure. In this ‘rest of the figure’, I don’t know whether the peptide opioids shown attached to the intracellular sides of the proto-receptor, ADR and MOR are supposed to represent the opioid themselves or the opioid-like sites on these proteins; the GSH-like compound on ADR is shown catalysing its reaction though it is bound to MORPH which is supposed to antagonise this catalysis (line 795)

FIGURE HAS BEEN MODIFIED TO SEPARATE OUT THE SECTIONS MORE OR LESS AS SUGGESTED. OPIOID-PEPTIDE-LIKE MODULES AND GSH-LIKE MODULES THAT HAVE BEEN INTEGRATED INTO THE RECEPTOR SEQUENCES HAVE BEEN RE-COLORED TO GRAY TO INDICATE THAT THEY ARE PART OF THE RECEPTOR (AND THIS IS NOW EXPLAINED IN THE FIGURE CAPTION). THE CATALYSIS ERROR HAS BEEN CORRECTED.

888 ‘One of the greatest mysteries that the mutation accumulation model leaves unresolved is why the random exploration of permutations does not lead to increasing loss of function over time and ever less integration within living systems when, instead, what is clearly observed across evolutionary time is an ever-increasing tendency toward systems integration exemplified in very stable interactomes. Selection for complementary modules can answer this conundrum’ and 902 ‘Interactome formation by means of swapping and selection for complementary modules produces, instead, a branched network of connections among evolutionary systems’ This is a powerful and persuasive idea. Perhaps supported by the work of Stuart Kauffman?

WE APPRECIATE THE SUPERFICIAL SIMILARITY BETWEEN OUR ORDERING MECHANISM AND KAUFFMAN’S BUT THERE IS ONE VERY IMPORTANT DIFFERENCE. KAUFFMAN’S MODELS OF EMERGENT ORDER ARE BASED ON RANDOM INTERACTIONS, ALL OF WHICH ARE EQUIVALENT, AMONG THE COMPONENTS OF HIS SYSTEMS. ONE OF US (R-B) HAS REPEATEDLY TRIED TO GET KAUFFMAN TO SEE THAT ADDING IN NON-RANDOM INTERACTIONS MEDIATED BY COMPLEMENTARITY WOULD MAKE HIS MODELS SIGNIFICANTLY MORE ROBUST BUT KAUFFMAN HAS REPEATEDLY REJECTED THE IDEA.  SO, YES, A MODIFIED FORM OF KAUFFMAN’S MODELS WOULD CERTAINLY YIELD THE OUTCOMES WE ARE PREDICTING HERE BUT THEY DO NOT INCORPORATE THE KEY IDEAS OF COMPLEMENTARITY AND MODULARITY THAT DRIVE OUR MECHANISM.  THIS WOULD MAKE A WONDERFUL FOLLOW-UP PAPER, HOWEVER!

Submission Date

08 October 2021

Date of this review

18 Oct 2021 17:09:30

Reviewer 2 Report

The topic proposed by the authors is interesting, consisting of a comprehensive meta-analysis about the co-evolution of opioid and adrenergic ligands and receptors as the start point for the novel engineering possibilities.

This article is written in a new format, which does not respect the classic sections of a scientific manuscript, however, this structure is innovative and the provided scientific information is easy to decipher.

Several aspects were found in this manuscript:

- the Figure 3 (line 353) must have the corresponding reference.

- for documentation, the authors used a big number of references, but the recent information in the field is missing.

- many of the references (972, 1017, 1207, ….., 1396) must be written according to the requirements of the journal,

- a careful check of the English language must be performed.

Author Response

REVIEWER 2:

Comments and Suggestions for Authors

The topic proposed by the authors is interesting, consisting of a comprehensive meta-analysis about the co-evolution of opioid and adrenergic ligands and receptors as the start point for the novel engineering possibilities.

This article is written in a new format, which does not respect the classic sections of a scientific manuscript, however, this structure is innovative and the provided scientific information is easy to decipher.

Several aspects were found in this manuscript:

- the Figure 3 (line 353) must have the corresponding reference.

ADDED

- for documentation, the authors used a big number of references, but the recent information in the field is missing.

THIS COMMENT IS BOTH TRUE AND NOT USEFUL. FOR EXAMPLE, THERE HAVE BEEN OVER 1000 PAPERS ON OPIOID RECEPTORS, 11 OF WHICH ADDRESS EVOLUTION, JUST THIS YEAR ACCORDING TO PUBMED. UNFORTUNATELY, NONE OF THESE ADDRESS ANY OF THE QUESTIONS POSED HERE.  THE CHOICE OF REFERENCES WAS MADE ON THE CRITERION THAT THE INFORMATION CONTAINED IN A REFERENCE DIRECTLY TESTED OR PROVIDED DATA CONCERNING ONE OF THE PREDICTIONS MADE BY THE MODEL ANALYZED IN THIS MANUSCRIPT. WE ARE NOT OF THE OPINION THAT JUST BECAUSE A PAPER IS MORE RECENT THAN ANOTHER THAT IT IS ANY MORE USEFUL OR APPROPRIATE THAN AN OLDER PAPER. MOREOVER, SINCE THE REVIEWER HAS NOT POINTED TO ANY PARTICULAR RECENT PAPERS AS BEING PARTICULARLY APPROPRIATE, WE HAVE NO WAY TO REDRESS THE ABSENCE OF ANY PARTICULAR REFERENCE. 

- many of the references (972, 1017, 1207, ….., 1396) must be written according to the requirements of the journal,

CORRECTED

- a careful check of the English language must be performed.

MANY AWKWARD SENTENCES AND SOME MISSPELLINGS HAVE BEEN CORRECTED PER REVIEWER 1’S DETAILED COMMENTS.

Submission Date

08 October 2021

Date of this review

20 Oct 2021 20:24:54

Reviewer 3 Report

In this study, the authors have proposed that opioid and adrenergic ligands and receptors co-evolved from a common set of modular precursors. The authors have demonstrated that the two receptor systems share molecularly complementary modules, which are highly conserved between them. The hypothesis is reasonable, and strongly supported by the former published experimental evidence and proteonomic homology comparation results. This could be very helpful for future study on human receptor functions. The study is very well designed, and the elements were clearly stated and understandable. I do not have criticisms and I am satisfied with the manuscript how it is.

Author Response

We thank the Reviewer for their very positive comments!